# Antidromic-rectifying gap junctions amplify chemical transmission at functionally mixed electrical-chemical synapses

Ping Liu[1], Bojun Chen[1], Roger Mailler[2] & Zhao-Wen Wang[1]

Neurons communicate through chemical synapses and electrical synapses (gap junctions). Although these two types of synapses often coexist between neurons, little is known about whether they interact, and whether any interactions between them are important to controlling synaptic strength and circuit functions. By studying chemical and electrical synapses between premotor interneurons (AVA) and downstream motor neurons (A-MNs) in the Caenorhabditis elegans escape circuit, we found that disrupting either the chemical or electrical synapses causes defective escape response. Gap junctions between AVA and A-MNs only allow antidromic current, but, curiously, disrupting them inhibits chemical transmission. In contrast, disrupting chemical synapses has no effect on the electrical coupling. These results demonstrate that gap junctions may serve as an amplifier of chemical transmission between neurons with both electrical and chemical synapses. The use of antidromic-rectifying gap junctions to amplify chemical transmission is potentially a conserved mechanism in circuit functions.

[1] Department of Neuroscience, University of Connecticut Health Center, Farmington, Connecticut 06030, USA. [2] Department of Computer Science, University of Tulsa, Tulsa, Oklahoma 74104, USA. Correspondence and requests for materials should be addressed to Z.-W.W. (email: zwwang@uchc.edu).

C hemical synapses and gap junctions (also known as electrical synapses) are the structural bases for neurons to communicate and to form functional circuits. Synaptic transmission at chemical synapses occurs through the release of neurotransmitters and binding to postsynaptic receptors, whereas at electrical synapses it occurs through direct current flow. While chemical and electrical synapses often coexist between neurons, they are generally conceived as independent transmission modalities. However, emerging evidence suggests that chemical and electrical synapses may interact to control synaptic strength and circuit function (see Pereda[1] for a review).

Our knowledge about interactions between chemical and electrical synapses has mostly come from studies of morphologically mixed synapses of invertebrates and lower vertebrates. At mixed synapses between auditory afferents and Mauthner cells in goldfish, excitatory chemical transmission is mediated by glutamate[2] while electrical coupling results from a heterotypic gap junction that is more conductive to antidromic than orthodromic current[3]. It has been proposed that the antidromic junctional current may promote cooperativity among afferents[3,4]. At mixed synapses between interneurons and motor neurons in zebrafish, motor neurons may recruit the interneurons and regulate their glutamate release through apparently non-rectifying gap junctions[5]. In Drosophila, mixed synapses between the giant fibre and tergotrochanteral motor neurons use acetylcholine for chemical transmission[6] and use heterotypic gap junctions formed by two different isoforms of the innexin Shaking-B for electrical coupling[7]. Analyses with the Xenopus oocyte heterologous expression system suggest that the gap junctions between giant fibre and tergotrochanteral motor neurons favour orthodromic current flow[7]. Mixed synapses are also abundant in mammalian brains[8–12], but very little is known about their functional properties and potential interactions.

The rather limited amount of knowledge of interactions between chemical and electrical synapses is mainly due to technical difficulties with performing paired current- and voltage-clamp recordings from neurons in situ, and a lack of genetic means to disrupt either the electrical or chemical component specifically. In Caenorhabditis elegans escape circuit, a pair of command interneurons (AVA) contact downstream A-type cholinergic motor neurons (A-MNs) through both chemical synapses and gap junctions. Although these chemical and electrical synapses are not closely associated in space (WormWiring, http://wormwiring.org/), they may functionally interact because neurons in C. elegans are essentially isopotential due to a very high membrane resistance, and changes in membrane voltage are instantaneously experienced by the entire neuron[13,14]. In this study, we took advantage of our recent success in performing paired voltage- and current-clamp recordings with C. elegans neurons and the high genetic amenability of the worm to investigate how electrical and chemical synapses interact to control synaptic transmission and escape behaviour. We found that chemical transmission from AVA to A-MNs is mediated by acetylcholine and an LGC-46 postsynaptic receptor, and that gap junctions between AVA and A-MNs only allow antidromic current. Importantly, we found that the rectifying gap junctions serve to amplify the excitatory chemical transmission from AVA to A-MNs and are critical to the escape behaviour. These findings show for the first time how chemical synapses and gap junctions may interplay to control a specific behaviour. The mechanisms uncovered here are potentially relevant to some morphologically or functionally mixed synapses in other systems, including mammals.

## Results

**Interneurons control A-MNs by producing PSC bursts**. The C. elegans locomotion neural circuit consists of 5 pairs of command interneurons and 58 ventral cord motor neurons[15]. Three pairs of the command interneurons (AVA, AVD and AVE) form synapses with A-type cholinergic MNs (A-MNs) mediating backward locomotion whereas the remaining two pairs (AVB and PVC) form synapses with B-type cholinergic MNs (B-MNs) mediating forward locomotion. Among the command interneurons in the backward neural circuit, AVA contacts A-MNs through both chemical synapses and gap junctions whereas AVD and AVE only through chemical synapses (Fig. 1a). To understand how command interneurons control A-MNs through chemical synapses, we started by analysing spontaneous postsynaptic currents (PSCs) recorded from VA5, which was chosen as a representative of A-MNs because it could be more easily identified based on its anatomical location than other A-MNs (Fig. 1b). We observed spontaneous PSCs occurring both sporadically and in bursts (Fig. 1c). In a histogram of decay time distribution, spontaneous PSCs may be divided into two distinct populations with a huge difference in the mean decay time ($0.92 \pm 0.03$ ms versus $59.25 \pm 3.43$ ms) (Fig. 1d). Spontaneous PSCs with a slow decay time (sPSCs) were also much larger in mean peak amplitude than those with a fast decay time (fPSCs) (Fig. 1d).

We defined PSC bursts as an apparent increase in PSC frequency accompanied by a persistent current lasting for $\geq 3$ s because we previously found that C. elegans motor neurons control body-wall muscle cells by producing PSC bursts, and that a burst duration of 3 s is close to the minimum to cause detectable changes in muscle activity[16]. The duration of PSC bursts in VA5 ranged from 3 to 35 s with the majority of them ($\sim 80\%$) lasting 5–15 s (Fig. 1e). The mean charge transfer rate during the burst was 6-fold higher than that outside the burst (Fig. 1e). Compared with extra-burst events, intra-burst PSCs showed significant increases in the frequency, mean amplitude and charge transfer but no change in the mean half width, 10–90% rise time, and decay time (Fig. 1f). These differences between extra- and intra-burst events are similar to those of spontaneous PSCs recorded from C. elegans body-wall muscle cells[16].

We wondered whether the PSC bursts observed in VA5 are important to the control of A-MNs by presynaptic neurons, as are those produced by MNs to the control of body-wall muscles[16]. To answer this question, we analysed temporal relationships between PSC bursts in VA5 and either PSC bursts or $Ca^{2+}$ transients in adjacent body-wall muscle cells, which are synchronous in activity[17]. Both PSC bursts and $Ca^{2+}$ transients in muscle cells are temporally correlated with PSC bursts in VA5 (Fig. 2 and Supplementary Movie 1). These observations suggest that command interneurons control A-MNs by causing PSC bursts.

**AVA interneurons control A-MNs by acetylcholine**. To identify the neurotransmitter(s) used by the command interneurons in the backward circuit, we began by testing the effects of various exogenous neurotransmitters on VA5 whole-cell current. In each experiment, a candidate exogenous neurotransmitter was applied to VA5 by pressure-ejection through a glass pipette. An inward current was expected upon the activation of either a non-selective cation channel or a $Cl^-$ channel because their equilibrium potentials were more depolarized than the holding potential ($-60$ mV) under our experimental conditions. We tested the effects of nine different candidate neurotransmitters, including acetylcholine, γ-aminobutyric acid (GABA), glutamate, aspartate, ATP, dopamine, octopamine, serotonin and tyramine. Only acetylcholine and GABA were able to induce a current (Supplementary Fig. 1a), suggesting the existence of both acetylcholine and GABA receptors in VA5. Nicotine and levamisole, which are agonists for some C. elegans acetylcholine receptors

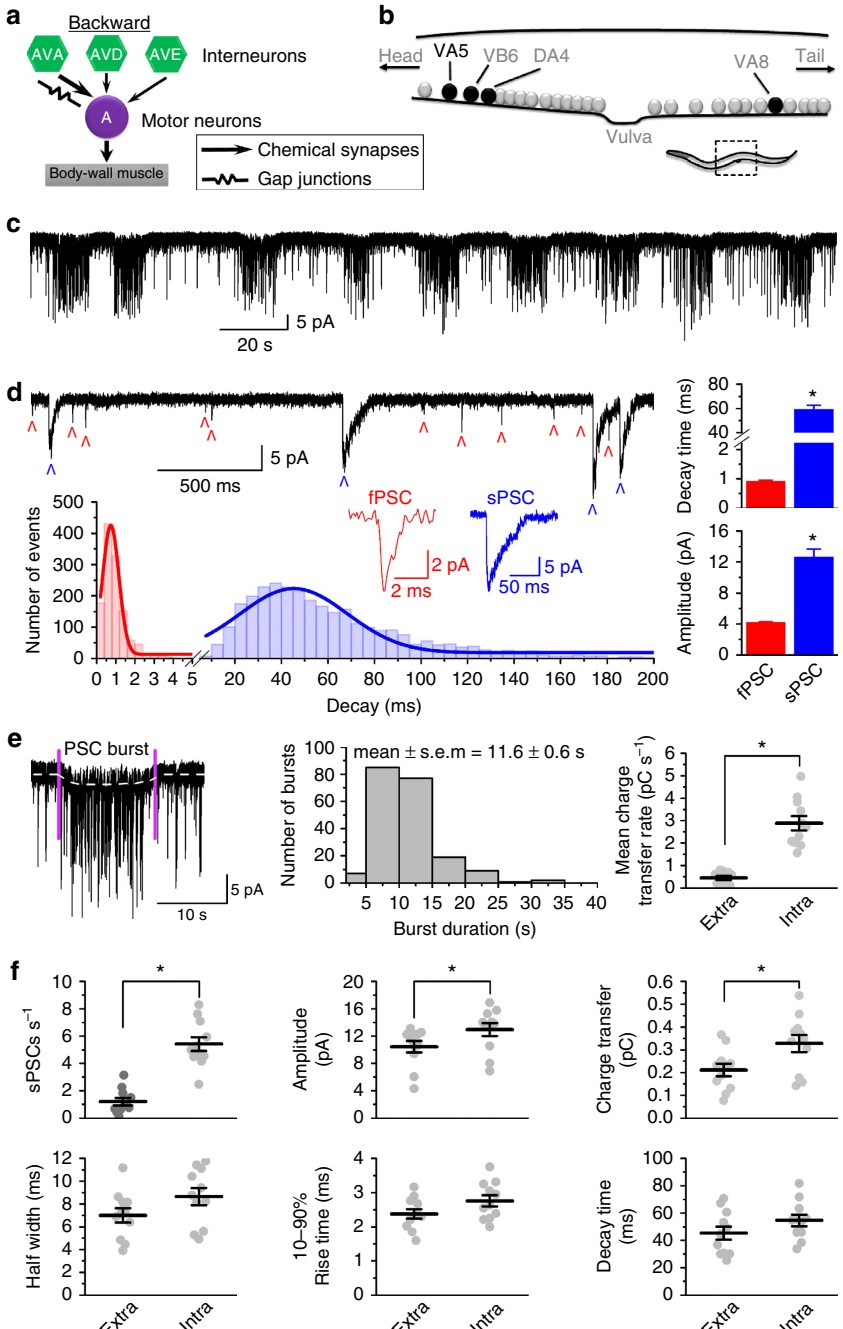

**Figure 1 | Properties of spontaneous postsynaptic currents (PSCs) recorded from VA5.** (**a**) Diagram showing synaptic connections between command interneurons (AVA, AVD and AVE) and A-type cholinergic motor neurons (A-MNs) in the *C. elegans* backward locomotion neural circuit. AVA contacts A-MNs through both chemical synapses and gap junctions whereas AVD and AVE contacts A-MNs through only chemical synapses. (**b**) Diagram showing the locations of VA5, VB6, DA4 and VA8. A portion of the whole worm (marked by a rectangle) is enlarged and shown. (**c**) A sample trace of spontaneous PSCs from VA5 showing PSC bursts and intervening sporadic events. (**d**) A segment of the trace from (**c**) displayed at a faster time scale to reveal fast-decaying PSCs (fPSCs, marked by red ˆ) and slow-decaying PSCs (sPSCs, marked by blue ˆ), representative fPSC and sPSC events, a histogram showing decay time distribution, and comparison of decay time and peak amplitude between fPSCs and sPSCs. (**e**) A sample PSC burst (between the two vertical purple lines), PSC burst duration distribution, and comparison of mean charge transfer rate between extra- and intra-burst periods. (**f**) Comparison of frequency, mean amplitude, charge transfer, half width, 10–90% rise time, and decay time between extra- and intra-burst PSC events. In **e,f**, each dot represents the averaged value of one experiment ($n = 11$). The asterisk (*) indicates a statistically significant difference ($P < 0.05$) based on either unpaired *t*-test (**d**) or paired *t*-test (**e,f**).

(AChRs)[18–21], were without an effect (Supplementary Fig. 1b). Choline, the precursor of acetylcholine, also showed no effect (Supplementary Fig. 1b).

Because chemical synaptic transmission from command interneurons to A-MNs presumably plays an excitatory role[15,22]

but GABA generally serves as an inhibitory neurotransmitter, we tested whether acetylcholine is a neurotransmitter used by the command interneurons. Indeed, d-tubocurarine, a competitive nicotinic AChR antagonist[23], abolished both spontaneous PSCs and exogenous acetylcholine-induced current in VA5

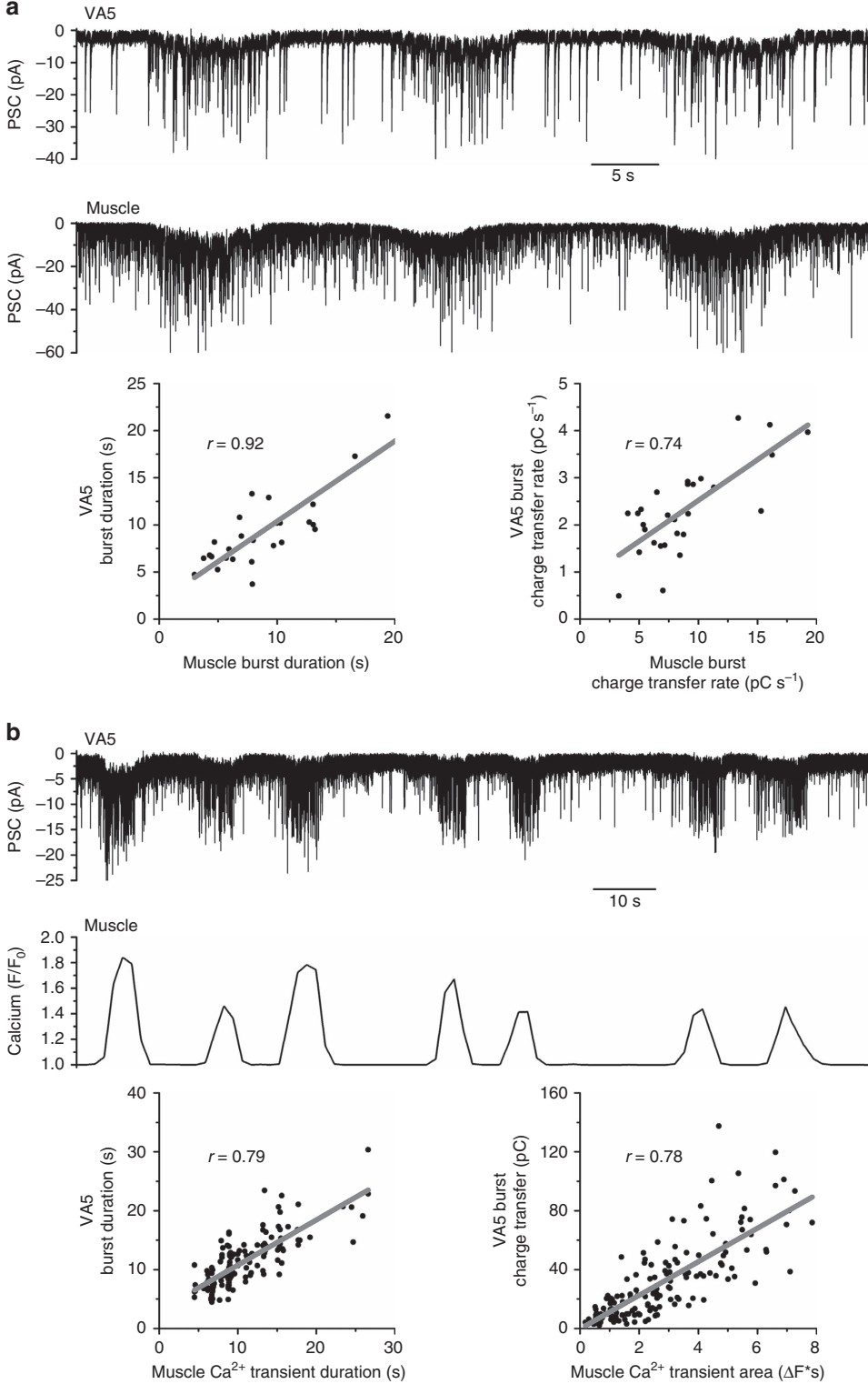

**Figure 2 | Postsynaptic current (PSC) bursts in VA5 are concurrent with PSC bursts and Ca$^{2+}$ transients in adjacent body-wall muscle cells.**
(**a**) Sample traces showing temporal relationship between PSC bursts recorded from VA5 and an adjacent muscle cell, and scatter plots showing correlations of duration and mean charge transfer rate between PSC bursts recorded from VA5 and muscle cells ($n = 5$). (**b**) Sample traces showing temporal relationship between PSC bursts recorded from VA5 and Ca$^{2+}$ transients from an adjacent muscle cell, and scatter plots showing correlations between their durations and strengths ($n = 10$). In both panels, each dot in a scatter plot represents one PSC burst or Ca$^{2+}$ transient.

(Supplementary Fig. 2a). Consistently, we found that CHA-1 (choline acetyltransferase) and UNC-17 (vesicular acetylcholine transporter), which are required for synthesizing acetylcholine and loading synaptic vesicles with acetylcholine, respectively[24], are important to spontaneous PSCs in VA5. In the hypomorphic mutants *cha-1(p1152)* and *unc-17(e245)* (nulls are nonviable)[24], the frequencies of sPSCs and fPSCs were greatly decreased (by 80–90%); and the remaining events of sPSCs but not fPSCs were

also much smaller in mean amplitude and charge transfer compared with wild type (Supplementary Fig. 2b). These observations suggest that spontaneous PSCs in VA5 result from acetylcholine release.

To determine whether command interneurons are a source of acetylcholine acting on A-MNs, we assessed the effect of d-tubocurarine on optogenetically evoked PSCs in VA5 using a strain expressing an integrated channelrhodopsin-2 (ChR2) transgene in command interneurons[16]. Each photostimulus caused a PSC burst associated with a large initial transient, which could be eliminated by d-tubocurarine (Fig. 3a). This observation suggests that command interneurons use acetylcholine to control A-MNs. Because AVA forms more chemical synapses with A-MNs than AVD and AVE[25,26] and is more important than the other two in backward locomotion[22,27–29], we tested whether AVA is a source of the acetylcholine. We first checked whether AVA expresses UNC-17, which is a marker of cholinergic neurons, by creating a transgenic strain in which GFP was expressed in cholinergic neurons under the control of P*unc-17* whereas mStrawberry specifically in the two AVA command interneurons using a *Cre-LoxP* recombination approach involving two promoters (P*flp-18* and P*gpa-14*)[30]. In transgenic worms, two neurons in the head region were colabelled by GFP and mStrawberry (Fig. 3b), suggesting that AVA is cholinergic. We then determined whether spontaneous PSCs in VA5 depend on acetylcholine release from AVA, and found that knockdown of *unc-17* specifically in AVA greatly inhibited PSC bursts and sPSCs but had no effect on fPSCs (Fig. 3c,d). The inhibition of sPSCs but not fPSCs in the AVA *unc-17* knockdown strain suggests that the RNA interference did not spread to A-MNs, which are the source of acetylcholine for fPSCs as described later. Finally, we determined whether suppressing AVA activity would inhibit spontaneous PSCs in VA5 using a strain expressing the histamine-activated $Cl^-$ channel HisCl1 specifically in AVA[31]. We found that histamine administration abolished PSC bursts and nearly eliminated sPSCs but did not affect fPSCs (Fig. 3e). Collectively, these observations suggest that PSC bursts and sPSCs in A-MNs arise from acetylcholine release from AVA, whereas fPSCs in A-MNs are due to acetylcholine release from a different source.

**LGC-46 AChR mediates AVA to A-MN transmission**. To identify the receptor(s) mediating AVA to A-MN transmission, we began by analysing spontaneous PSCs in VA5 in mutants of selected genes of the Cys-loop superfamily of ligand-gated ion channels[32]. *acr-5, acr-8, acr-14, acr-15, acr-18, acr-21, lgc-46* and *lgc-55* were chosen for the analysis because they are known to be expressed in at least some of the ventral cord motor neurons (www.wormbase.org) but are not components of a previously identified motor neurons extrasynaptic receptor[20,33]. We found that mutations of *lgc-46* (Fig. 4a) but not the other genes (Supplementary Fig. 3) caused a great reduction ($\sim$70%) in the frequency of sPSCs with the remaining events having a smaller mean amplitude and less charge transfer compared with wild type. Knockdown of *lgc-46* specifically in A-MNs also inhibited sPSCs (Fig. 4a). In contrast, fPSCs were unchanged in the mutants compared with wild type (Fig. 4a). The deficiencies of sPSCs in *lgc-46* mutants could be rescued by expressing wild-type LGC-46 specifically in A-MNs (Fig. 4a). These observations suggest that LGC-46 is a key component of the AChR mediating sPSCs in A-MNs.

Although expression of *lgc-46* (also known as Y71D11A.5) in ventral cord MNs has been reported[34], A-MNs were not among those identified in the previous study. To confirm that *lgc-46* is expressed in A-MNs, we created a transgenic strain coexpressing

GFP and mStrawberry under the control of P*lgc-46* and the A-MN-specific promoter P*unc-4*, respectively. In the transgenic worms, GFP expression was observed in ventral cord MNs, many head neurons, and some neurons in the tail (Fig. 4b). All mStrawberry-positive neurons were colabelled by GFP, suggesting that *lgc-46* is expressed in A-MNs. In addition, GFP was observed in some ventral cord motor neurons not labelled by mStrawberry, which could be GABAergic MNs that reportedly express *lgc-46* (ref. 34).

To confirm that LGC-46 may form an ionotropic AChR, we expressed LGC-46 in *Xenopus* oocytes and measured whole-cell current using the two-microelectrode voltage clamp technique. Acetylcholine caused inward current in a concentration-dependent manner with the half maximal effective concentration ($EC_{50}$) being $50.6 \pm 2.6\,\mu M$ (Fig. 4c). In contrast, none of the other tested agents, including levamisole, nicotine, choline, and glycine, showed any effect on the whole-cell current (Fig. 4d). These observations suggest that LGC-46 may form an ionotropic receptor that is specifically activated by acetylcholine.

**ACR-14 autoreceptor in A-MNs facilitates transmitter release**. To identify the AChR mediating fPSCs in VA5, we scrutinized recording traces of spontaneous PSCs from the analysed mutants. We found that *acr-14* is important to fPSCs. In the loss-of-function mutant *acr-14(ok1155)*[34], fPSCs were decreased by $\sim$80% in frequency with the remaining events having a smaller mean amplitude and less charge transfer compared with wild type (Fig. 5a). In contrast, sPSCs were normal in the mutant (Fig. 5a). The deficiencies of fPSCs could be rescued by expressing wild-type ACR-14 specifically in A-MNs (Fig. 5a). These observations suggest that ACR-14 is a key component of the AChR mediating fPSCs.

We then tried to identify the source of acetylcholine causing fPSCs in VA5. A previous study showed that GABA miniature PSCs (minis) in interneurons of the molecular layer of rat cerebellum include large-amplitude 'ordinary minis' and small-amplitude 'preminis' (presynaptic miniature currents); and that the latter result from the activation of autoreceptors[35]. We wondered whether fPSCs in VA5 result from a similar mechanism. To explore this possibility, we first examined the effect of *unc-17* knockdown specifically in A-MNs, and observed a significant reduction in the frequency of fPSCs but not sPSCs (Fig. 5a), suggesting that fPSCs depend on acetylcholine release from A-MNs. The inhibition of fPSCs but not sPSCs in the A-MN-specific *unc-17* knockdown strain suggests that the RNA interference did not spread to AVA, which is the source of acetylcholine for sPSCs as shown earlier (see Fig. 3). We then tested whether inhibiting neurotransmitter release from the same neuron (VA5) may inhibit fPSCs by including 5 mM BAPTA in the pipette solution used for whole-cell recording. BAPTA is a membrane impermeant fast $Ca^{2+}$ chelator[36], and is expected to inhibit neurotransmitter release from VA5 through depleting free $Ca^{2+}$ in the cytosol. A great decrease in fPSC frequency was observed in the BAPTA-treated VA5 (Fig. 5a). Neither the A-MN-specific *unc-17* knockdown nor the BAPTA treatment affected sPSCs (Fig. 5a). These observations suggest that fPSCs in A-MNs likely result from the activation of an autoreceptor containing ACR-14.

To confirm that *acr-14* is expressed in A-MNs, We coexpressed GFP and mStrawberry under the control of P*acr-14* and the A-MN-specific promoter P*unc-4* (ref. 37), respectively. In the transgenic worms, GFP was observed in all mStrawberry-labelled neurons plus some other neurons along the ventral nerve cord (Fig. 5b), suggesting that *acr-14* is expressed in A-MNs as well as some other MNs, which could be D-MNs and B- and AS-

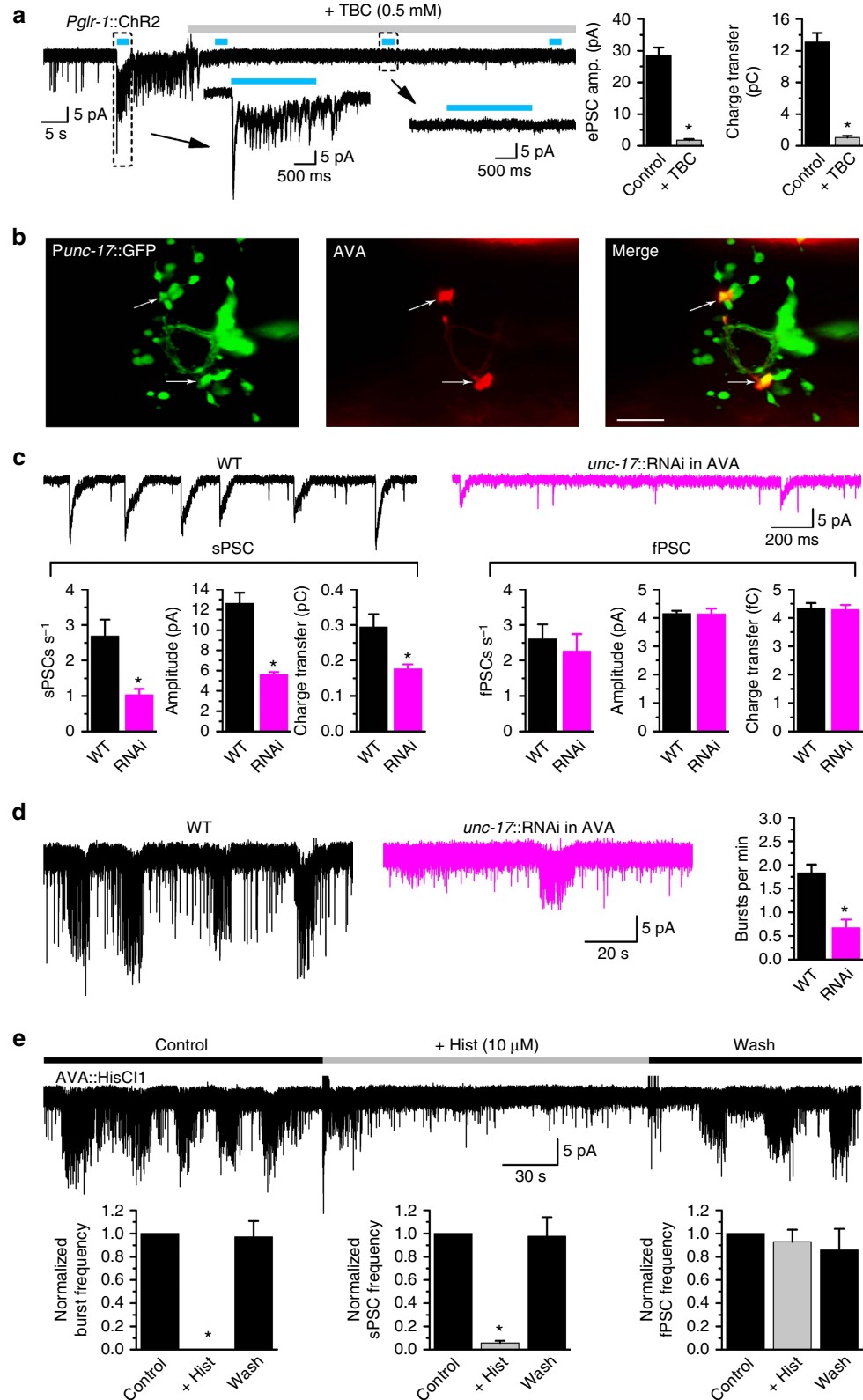

**Figure 3 | AVA command interneurons are cholinergic neurons responsible for slow postsynaptic currents (sPSCs) in VA5.** (**a**) d-Tubocurarine (TBC) inhibits both spontaneous PSCs and optogenetically evoked PSCs (ePSCs) in a strain expressing channelrhodopsin-2 (ChR2) in command interneurons under the control of P*glr-1*. The bar graphs show comparisons of ePSC peak amplitude and total charge transfer during the 2-s light pulse. (**b**) The two AVA command interneurons (labelled by mStrawberry) are colabelled by GFP, which is expressed in cholinergic neurons under the control of P*unc-17*. Scale bar, 10 μm. (**c,d**) Knockdown (RNAi) of *unc-17* specifically in AVA inhibits sPSCs and PSC bursts but has no effect on fast PSCs (fPSCs) compared with wild type (WT). (**e**) Application of histamine to a strain expressing HisCl specifically in AVA essentially eliminates PSC bursts and sPSCs but does not affect fPSCs. The asterisk (*) indicates a statistically significant difference ($P < 0.05$) compared with either WT or the Control period (paired *t*-test in **a**, unpaired *t*-test in **c,d**, and one-way ANOVA with Tukey's post hoc test in **e**). Sample size (*n*) was 8 in **a**, 11 WT and 8 RNAi in **c**, 12 WT and 8 RNAi in **d**, and 8 in **e**.

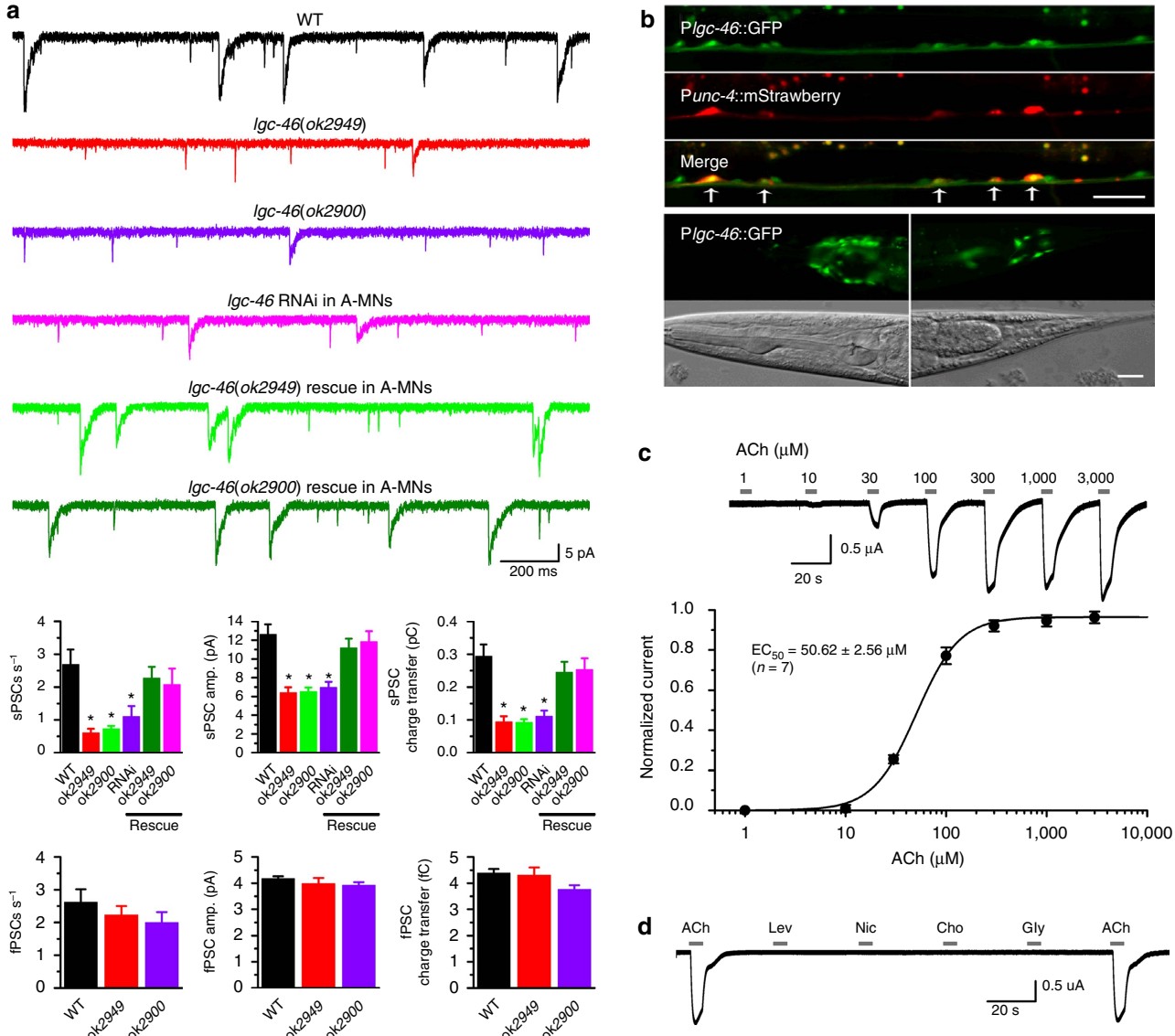

**Figure 4 | Slow postsynaptic currents (sPSCs) but not fast PSCs (fSPCs) in VA5 result from the activation of an LGC-46 acetylcholine receptor.**
(**a**) Sample traces and statistical comparisons of spontaneous PSCs among wild type (WT) ($n = 11$), lgc-46(ok2949) ($n = 11$), lgc-46(ok2900) ($n = 6$), lgc-46 knockdown (RNAi) specifically in A-type cholinergic motor neurons (A-MNs) ($n = 11$), and the two mutants rescued by expressing wild-type LGC-46 specifically in A-MNs ($n = 10$ ok2949; $n = 7$ ok2900). The asterisk (*) indicates a statistically significant difference compared with WT ($P < 0.05$, one-way ANOVA with Tukey's post hoc test). ok2949 and ok2900 are deletion mutants involving one or two exons (www.wormbase.org). (**b**) In a strain coexpressing GFP under the control of lgc-46 promoter and mStrawberry under the control of unc-4 promoter, all A-type cholinergic motor neurons are colabelled by the two fluorescent proteins (top three panels). The smaller fluorescent puncta (mostly away from the ventral nerve cord) are due to autofluorescence from the gut. GFP is also expressed in many neurons in the head and tail (bottom panels). Scale bar, 20 μm. (**c**) A sample trace showing whole-cell current of an oocyte in response to increasing concentrations of acetylcholine (top), and normalized acetylcholine concentration-response curve (bottom). Data are shown as mean ± s.e. (**d**) A sample trace showing that inward current is induced by acetylcholine but not by levamisole, nicotine, choline, or glycine (all at 100 μM). Holding voltage was −60 mV in all experiments.

type MNs reported in previous studies[34,38,39]. In addition, GFP expression was observed in several head and tail neurons (Fig. 5b).

What could be the role of the ACR-14 autoreceptor in motor neuron function? To answer this question, we compared the frequency and mean amplitude of spontaneous PSCs as well as the amplitude of evoked PSCs at the neuromuscular junction between wild type and acr-14(ok1155). The mutant showed ∼30% decrease in both the amplitude of evoked PSCs and the frequency of spontaneous PSCs compared with wild type without a change in the mean amplitude of spontaneous PSCs (Fig. 5c).

These observations suggest that a physiological role of the ACR-14 autoreceptor is to facilitate spontaneous and evoked neurotransmitter release.

**Gap junctions between AVA and A-MN are antidromic rectifiers.** We analysed properties of the gap junctions between AVA interneurons and A-MNs by performing paired voltage- and current-clamp recordings with VA5 and an AVA (Fig. 6a). We found that gap junctions between them are strongly rectifying in the AVA direction. Significant junctional current ($I_j$) is observed only when the membrane voltage ($V_m$) of VA5 is more

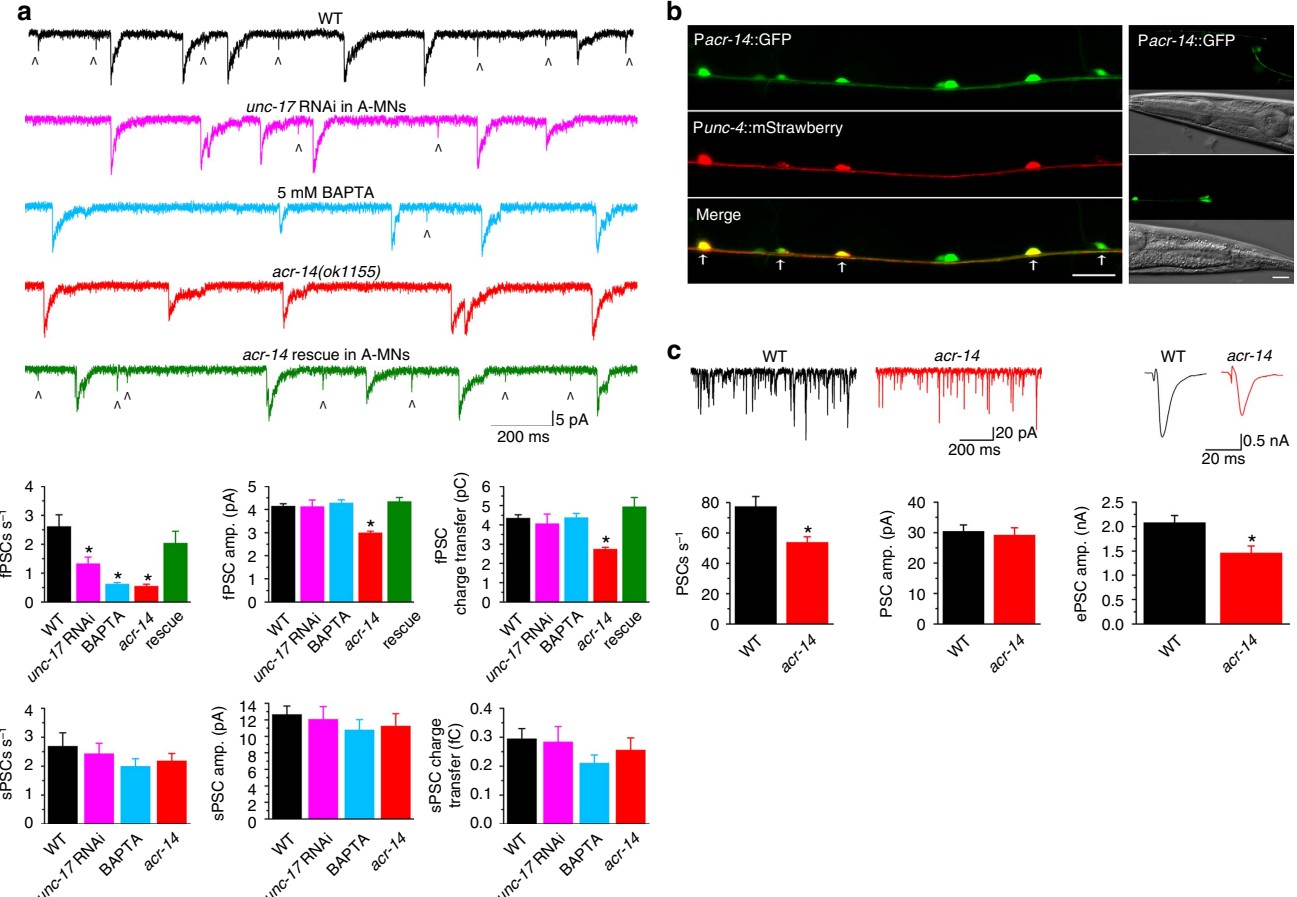

**Figure 5 | ACR-14 acts as an autoreceptor to mediate fast PSCs (fPSCs) in VA5 and enhances spontaneous and evoked neurotransmitter release.**
(**a**) Sample traces of spontaneous PSCs and statistical comparisons among wild type (WT), *unc-17* knockdown (RNAi) specifically in A-type cholinergic motor neurons (A-MNs), WT recorded with BAPTA in the pipette solution, *acr-14(ok1155)*, and *acr-14(ok1155)* rescued specifically in A-MNs by expressing wild-type ACR-14 under the control of P*unc-4*. fPSCs are marked by the caret sign (^). The sample size (*n*) was 11 WT, 7 *unc-17* RNAi, 5 BAPTA, 9 *acr-14(ok1155)*, and 7 *acr-14(ok1155)* rescue. (**b**) In a strain coexpressing GFP under the control of P*acr-14* and mStrawberry under the control of P*unc-4*, all A-MNs are colabelled by the two fluorescent proteins (left). GFP expression was also observed in a few head (right top) and tail (right bottom) neurons in this strain. Corresponding differential interference contrast images are shown below. Scale bar, 20 μm. (**c**) The frequency of spontaneous PSCs and the amplitude of evoked PSCs recorded from body-wall muscle cells are decreased while the mean amplitude of spontaneous PSCs has no change in *acr-14(ok1155)* (*n* = 15) compared with WT (*n* = 14). The asterisk (*) indicates a statistically significant difference compared with WT based on either one-way *ANOVA* with Tukey's post hoc test (**a**) or unpaired *t*-test (**c**).

depolarized than that of AVA (Fig. 6b), indicating that $I_j$ flows only in the AVA direction. Consistently, AVA showed large $V_m$ changes in response to current injections into VA5 whereas VA5 showed little $V_m$ change in response to current injections into AVA, as quantified by the coupling coefficient (CC) (Fig. 6c). Cell-specific rescue experiments for mutant locomotion defects have led to the suggestion that the gap junctions between AVA and A-MNs are heterotypic gap junctions formed by UNC-7 in AVA and UNC-9 in A-MNs[22,40] but this conclusion remains to be substantiated by electrophysiological evidence. We found that, in both *unc-7* and *unc-9* mutants, $I_j$ between AVA and VA5 was essentially absent, and CC was very small even when current was injected into VA5 (Fig. 6b,c), which confirmed the importance of UNC-7 and UNC-9 in the electrical coupling.

We also analysed electrical coupling between AVA and VA8, which is another A-MN innervating ventral muscles but is located far away from VA5 (Fig. 1b). VA8 was chosen because it was analysed in a previous study of electrical and chemical transmission between AVA and A-MNs[22]. We found that $I_j$ is similarly rectifying in the AVA direction (Supplementary Fig. 4). The observations of rectifying gap junctions between AVA and

two A-MNs from different locations suggest that gap junctions between AVA and A-MNs are likely antidromic rectifiers.

**Electrical coupling amplifies chemical transmission.** To determine how gap junctions might interact with chemical synapses between AVA and A-MNs, we compared spontaneous PSCs in VA5 between wild type and two strains with either *unc-7* knockdown specifically in AVA or *unc-9* knockdown specifically in A-MNs. Both knockdown strains showed little electrical coupling between AVA and VA5 (Fig. 7a,b). Remarkably, the frequency of PSC bursts was greatly reduced in the two knockdown strains compared with wild type (Fig. 7c), suggesting that the rectifying gap junctions normally serve as an amplifier of chemical transmission from AVA to A-MNs.

To test whether the observed inhibition of chemical synaptic transmission in the *unc-7* knockdown strain resulted from a secondary developmental defect, we examined the effects of conditional UNC-7 depletion on synaptic transmission between AVA and A-MNs using an auxin-inducible degradation approach[41]. We created a strain in which UNC-7 is tagged with the degron at the C-terminus by Crispr/Cas9-based genome

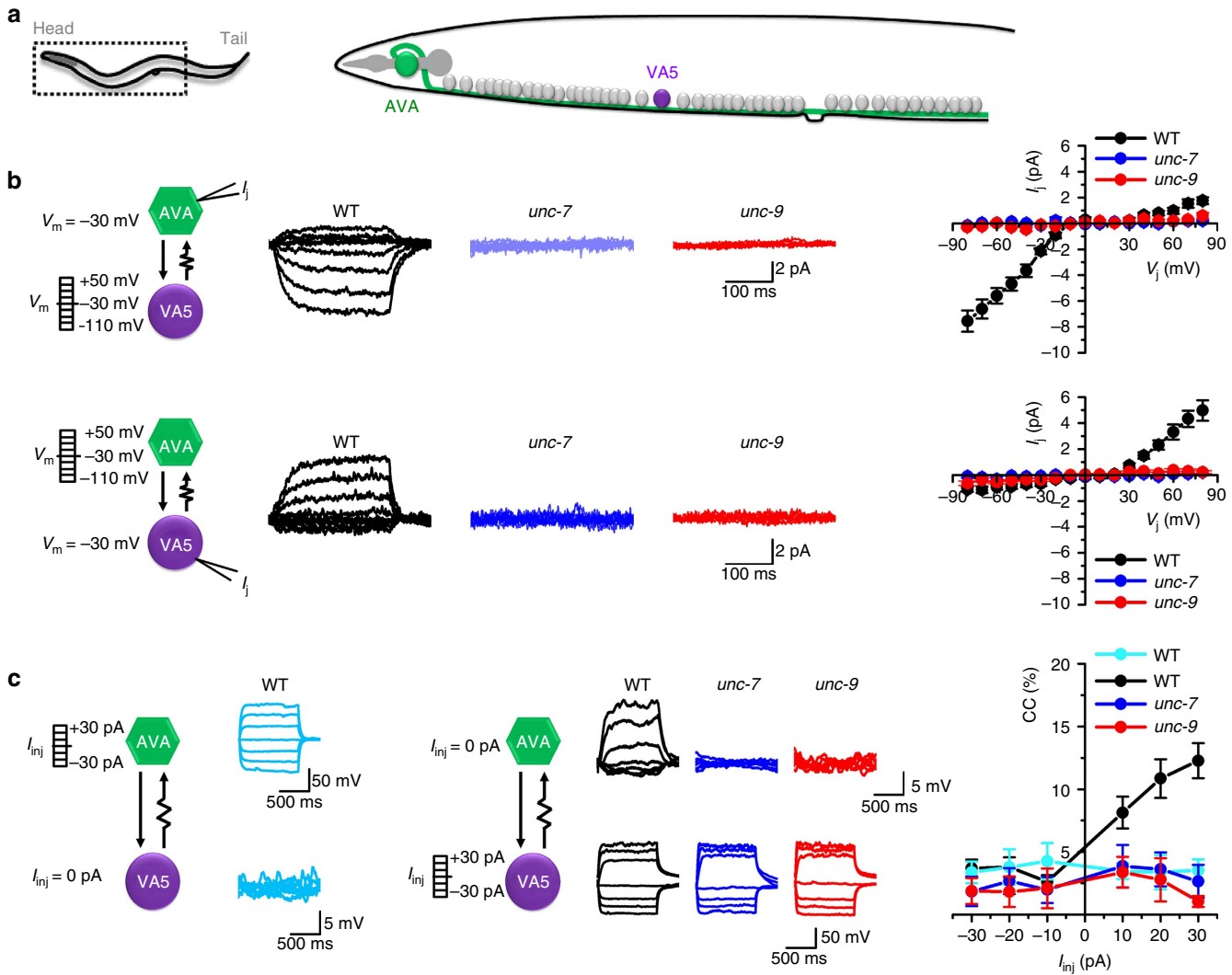

**Figure 6 | Gap junctions between AVA and VA5 only allow antidromic junctional current ($I_j$) and require the innexins UNC-7 and UNC-9 to function.**
(**a**) Diagram showing the locations of AVA and VA5. (**b**) $I_j$ between AVA and VA5, strongly rectifying in the AVA direction, is present in wild type (WT, $n = 10$) but not in either *unc-7(e5)* ($n = 8$) or *unc-9(fc16)* ($n = 8$). $I_j$ was recorded from one neuron held constantly at $-30$ mV (neuron #1) while the other neuron (neuron #2) was stepped to various membrane voltage ($V_m$) from a holding voltage of $-30$ mV. $V_j = V_m$ of neuron #1 $- V_m$ of neuron #2. Whole-cell membrane currents of the neuron receiving the voltage steps are not shown for clarity. (**c**) Current injection ($I_{inj}$) into VA5 but not AVA causes significant depolarization of the uninjected neuron in WT ($n = 10$). $I_{inj}$ into VA5 has little effect on AVA in the *unc-7* ($n = 8$) and *unc-9* ($n = 8$) mutants. Coupling coefficient (CC) is the ratio of $V_m$ changes of the uninjected neuron over the injected neuron.

editing[42] and the TIR1 F-box protein is expressed specifically in AVA from an extrachromosome array. TIR1 interacts with a number of endogenous proteins to form a functional E3 ubiquitin ligase and, in the presence of auxin, binds the degron-tagged protein and triggers its degradation. Auxin administration to this transgenic strain abolished $I_j$ between AVA and VA5, and greatly inhibited PSC bursts in VA5 (Supplementary Fig. 5), which are similar to the effects of AVA-specific *unc-7* knockdown. These observations reinforce the notion that the antidromic $I_j$ between AVA and A-MNs facilitates chemical synaptic transmission.

We also analysed the effects of AVA-specific *unc-7* knockdown on electrical coupling between AVA and DA4 (a representative A-MN innervating dorsal muscles), and on PSC bursts in DA4. We found that $I_j$ is abolished, and PSC bursts are greatly inhibited in the knockdown strain (Supplementary Fig. 6), suggesting that gap junctions between AVA and A-MNs innervating dorsal muscles play a similar regulatory role in chemical transmission.

We then examined whether chemical transmission may regulate electrical coupling between AVA and A-MNs. To this end, we analysed $I_j$ current between AVA and VA5 in the AVA-specific *unc-17* knockdown strain described earlier (Fig. 3). We found that $I_j$ was indistinguishable between wild type and the knockdown strain (Supplementary Fig. 7), suggesting that a deficiency of chemical transmission between AVA and A-MNs does not compromise electrical coupling.

Given that gap junctions and chemical synapses between AVA and VA5 are not physically close, the observed functional interactions between them suggest that these neurons likely have very high membrane resistance. We therefore measured input resistance, which almost completely results from the membrane resistance, in AVA and VA5 at holding voltages close to their resting membrane potentials (AVA $-25$ mV; VA5 $-70$ mV)[43,44]. The measured input resistance was $2.50 \pm 0.19$ GΩ for AVA ($n = 9$) and $3.47 \pm 0.18$ GΩ for VA5 ($n = 22$). These unusually high input resistance values suggest

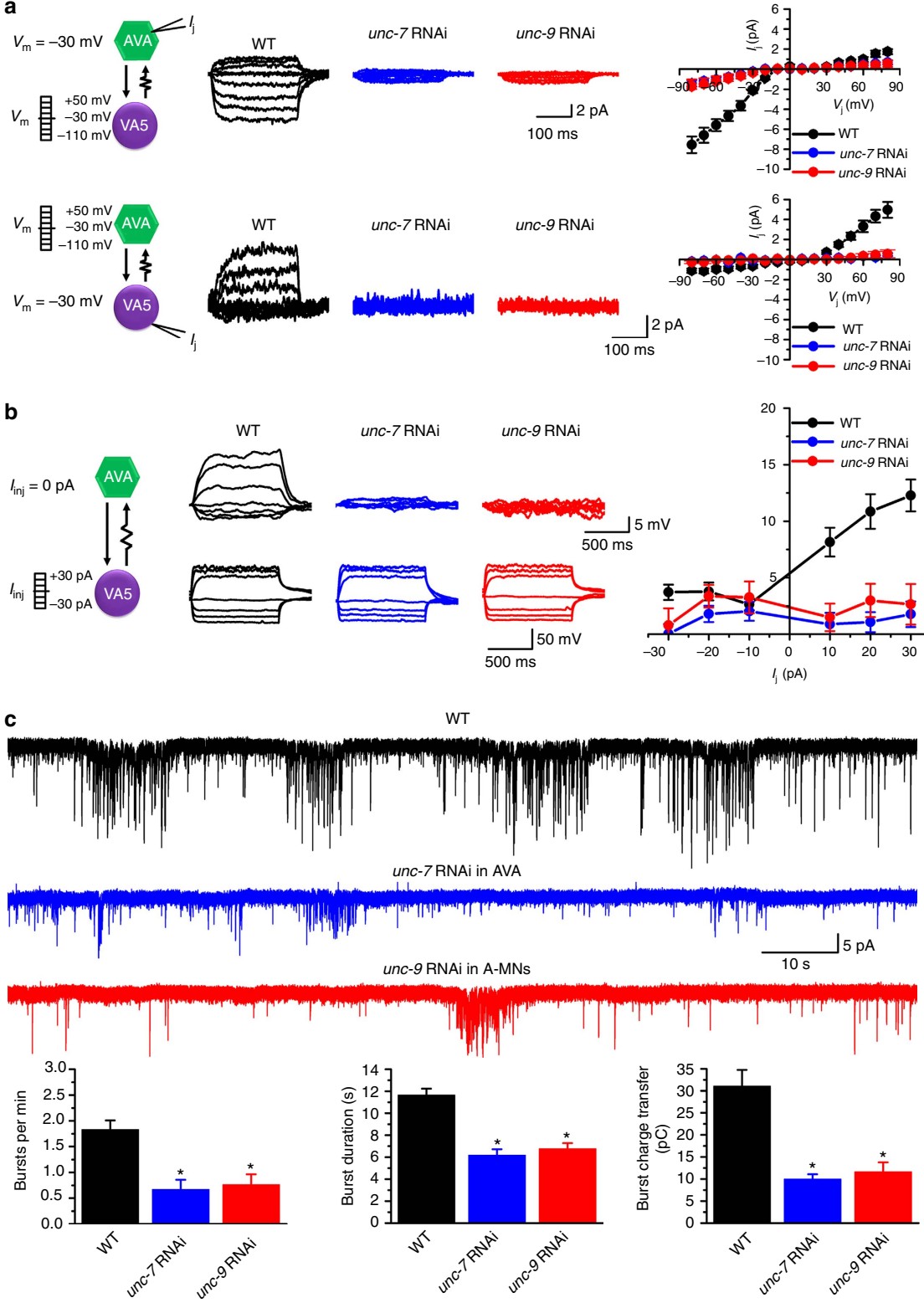

**Figure 7 | Deficient electrical coupling inhibits chemical synaptic transmission between AVA and VA5. (a)** Knockdown of either *unc-7* specifically in AVA or *unc-9* specifically in A-type cholinergic motor neurons essentially abolishes junctional current ($I_j$) between AVA and VA5. $I_j$ was recorded from one neuron held constantly at −30 mV (neuron #1) while the other neuron (neuron #2) was stepped to various membrane voltage ($V_m$) from a holding voltage of −30 mV. $V_j = V_m$ of neuron #1 − $V_m$ of neuron #2. Whole-cell membrane currents of the neuron receiving the voltage steps are not shown for clarity. **(b)** Current injections ($I_{inj}$) into VA5 cause $V_m$ changes in AVA of wild type but not of either the *unc-7* or the *unc-9* knockdown strain. In both **a,b**, *n* was 10 wild type (WT), 7 *unc-7* RNAi and 8 *unc-9* RNAi. **(c)** PSC bursts recorded from VA5 are greatly inhibited in either the *unc-7* ($n = 7$) or *unc-9* ($n = 7$) knockdown strain compared with WT ($n = 12$). Data are shown as mean ± s.e. The asterisk (*) indicates a statistically significant difference compared with WT ($P < 0.05$, one-way ANOVA followed by Tukey's HSD test).

that a small $I_j$ could have a major impact on the membrane voltage in these neurons, and that a change in membrane voltage may instantaneously spread to the entire neuron.

**Escape depends on both chemical and electrical synapses.** The identification of the key molecules for chemical and electrical transmission between AVA and A-MNs made it possible to determine the roles of these two synapses in *C. elegans* escape response. We specifically disrupted either the chemical or the electrical transmission, and analysed their effects on an escape response to a mechanical touch on the head. In response to the mechanical touch, worms move backward to escape. We counted the number of dorsoventral tail swings of each worm in response to the nose touch and used it as a measure of the backward locomotion. Compared with wild type, backward locomotion is severely compromised in the AVA-specific *unc-17* knockdown strain, the *lgc-46* mutants, and the A-MN-specific *lgc-46* knockdown strain; and the *lgc-46* mutant locomotion defect can be rescued completely by expressing wild-type LGC-46 under the control of its native promoter (Fig. 8a; Supplementary Movies 2–8), suggesting that chemical synaptic transmission between AVA and A-MNs is important to backward locomotion. Backward locomotion was also severely defective in the AVA-specific *unc-7* knockdown and the A-MN-specific *unc-9* knockdown strains (Fig. 8a; Supplementary Movies 9 and 10), which were used in our electrophysiological analyses (Fig. 7). Since ectopic gap junctions between AVA and B-MNs have been observed in *unc-7* mutant[45], we tested whether such ectopic gap junctions also occur in the AVA-specific *unc-7* knockdown strain by analysing the electrical coupling between AVA and VB6, a representative of B-MNs. We did not observe any $I_j$ in either wild type or the *unc-7* knockdown strain (Supplementary Fig. 8). Taken together, we have demonstrated at both the synaptic and behavioural levels that gap junctions between AVA and A-MNs serve to amplify chemical synaptic transmission from AVA to A-MNs, as illustrated in the model diagram (Fig. 8b).

## Discussion

Our work with *C. elegans* shows that antidromic rectifying gap junctions can facilitate orthodromic chemical transmission between neurons with mixed transmission modalities, and that this functional interaction is important in controlling a specific behaviour. Our study also shows that motor neurons are not simply passive responders to commands but may modulate the activities of higher order neurons retrogradely.

Gap junctions are formed by connexins in vertebrates and innexins in invertebrates. Each gap junction consists of two hemichannels that are either identical or different in composition. Gap junctions of identical hemichannels are known as homotypic gap junctions while those of different hemichannels are known as heterotypic gap junctions. Among gap junctions in native tissues studied to date, the majority allow current flow in both directions equally, while only a small number pass current preferentially in one direction. Rectification is generally a property of heterotypic gap junctions formed by either two different gap junction proteins (connexins or innexins) or two different isoforms of one gap junction protein[3,7,46]. In this study, we showed that gap junctions between AVA and A-MNs allow current flow only in one direction, which is probably the strongest degree of rectification ever observed with gap junctions in native tissues. Our analyses suggest that this antidromically rectifying junctional current serves to amplify chemical synaptic transmission.

Among a total of 25 innexin genes in *C. elegans*, only mutants of *unc-7* and *unc-9* are associated with obvious locomotion defects. Cell-specific rescue experiments of the locomotion defects

have led to the suggestion that the gap junctions between AVA and A-MNs are likely heterotypic gap junctions formed by UNC-7 in AVA and UNC-9 in A-MNs[22,40]. Our electrophysiological analyses confirmed this prediction. An earlier study found that AVA is hyperactive in *unc-7* mutant, and that rescuing electrical coupling between AVA and A-MNs in *unc-7* mutant facilitates forward instead of backward locomotion. Based on these observations, it was concluded that the electrical coupling between AVA and A-MNs normally inhibits backward locomotion by shunting excitatory current away from AVA into A-MNs[22]. In contrast, our results indicate that the gap junctions between AVA and A-MNs only allow current flow from A-MNs into AVA, and that a deficiency of the electrical coupling compromises rather than facilitates backward locomotion. Because ectopic gap junctions between AVA and B-MNs have been observed in *unc-7* mutant[45], and because B-MNs are important to forward locomotion[15], it is unclear whether the observations of the previous study[22] have been complicated by the ectopic gap junctions. Alternatively, other differences in experimental approaches could have contributed to the different conclusions. In the present study, we showed that AVA-specific removal of UNC-7 in adult worms produced similar results as AVA-specific *unc-7* RNAi. Furthermore, we did not observe any electrical coupling between AVA and a representative B-MN in the *unc-7* RNAi strain. Therefore, the effects of electrical coupling on chemical transmission and locomotion behaviour observed in the *unc-7* RNAi strain are unlikely to be caused by either ectopic gap junctions or developmental defects.

The identity of neurotransmitter used by AVA has been uncertain. Studies based on either expressing GFP under the control of specific promoters or immunostaining against specific neuron markers have led to controversial conclusions. One study suggests that the neurotransmitter is acetylcholine[47] whereas another suggests that it is glutamate[48]. The possibility of acetylcholine being the neurotransmitter is also disputed by results of two other studies[33,49]. Although the approaches used in the previous studies are generally effective in reporting gene expression patterns, there are instances where immunoreactivity is undetected because of either a low protein expression level or a low affinity of the antibody, and where the GFP reporter cannot accurately recapitulate the native expression pattern of a gene because of the absence of either introns or distant regulatory elements important to gene expression in the *gfp* transgene[50–52]. The present study confirms that acetylcholine is the neurotransmitter used by AVA. In addition, we identified LGC-46 as a key subunit of the postsynaptic AChR mediating synaptic transmission from AVA. The identification of both the neurotransmitter and a key component of the postsynaptic receptor for chemical transmission from AVA to A-MNs may serve as an important foundation for future studies on *C. elegans* locomotion neural circuit, which is commonly used as a model to understand how a small number of neurons may produce complex locomotion behaviours.

*C. elegans* neurons do not fire all-or-none overshooting action potentials[13]. It is generally thought that *C. elegans* neurons perform their physiological functions by producing graded potentials but it is not totally clear how graded potentials of presynaptic neurons may alter the activity of postsynaptic neurons. Our earlier study shows that *C. elegans* motor neurons control body-wall muscle cells by producing PSC bursts[16]. It is difficult to generalize this finding to other synapses without experimental evidence because body-wall muscle cells differ from neurons in that they fire all-or-none overshooting action potentials[53,54]. The findings that both AVA command interneurons and motor neurons control postsynaptic cells by producing PSC bursts suggest that PSC

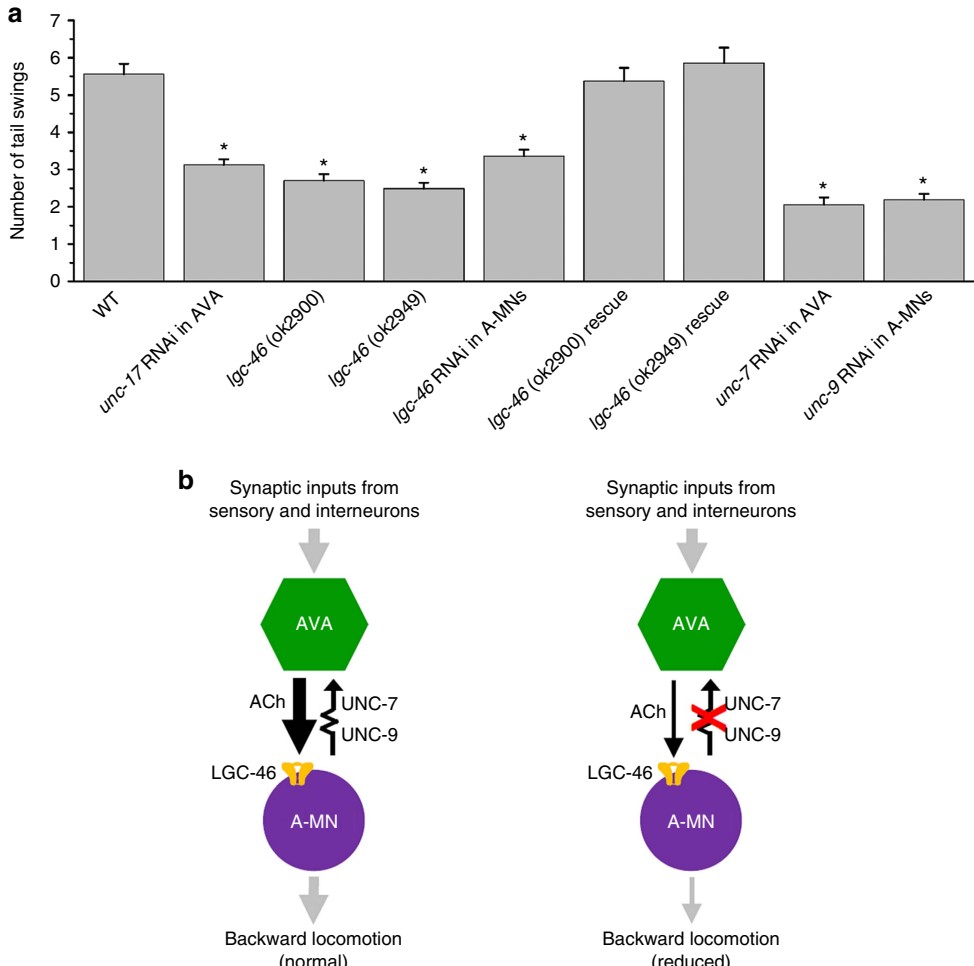

**Figure 8 | Chemical and electrical synapses between AVA and A-type cholinergic motor neurons (A-MNs) interact to control backward locomotion.**
(**a**) Backward locomotion is impaired when either the chemical or the electrical transmission is deficient. The number of backward locomotion-related tail swings in response to a head touch was compared among wild type (WT), *unc-17* knockdown specifically in AVA, *lgc-46* mutants, *lgc-46* knockdown specifically in A-MNs, *lgc-46* mutants rescued by expressing wild-type LGC-46 under the control of P*lgc-46*, *unc-7* knockdown specifically in AVA, and *unc-9* knockdown specifically in A-MNs. *n* was 100 for all groups. Data are shown as mean ± s.e. The asterisk (*) indicates a statistically significant difference compared with WT ($P < 0.05$, one-way ANOVA followed by Tukey's HSD test). (**b**) Diagrams showing how chemical transmission occurs between AVA and A-MNs, and how gap junctions contribute to the chemical transmission. AVA excites A-MNs by releasing acetylcholine and activating a postsynaptic receptor containing LGC-46. Heterotypic gap junctions formed by UNC-7 in AVA and UNC-9 in A-MNs are strongly rectifying in the AVA direction. Chemical transmission is strong in wild type due to antidromic junctional current (left) but weak when the electrical coupling is disrupted (right).

bursts might be generally used for signal transduction at chemical synapses in *C. elegans*.

ACR-14 has not been implicated in the function of A-MNs although it has been speculated as an autoreceptor[39]. Our results suggest that ACR-14 in A-MNs is indeed an autoreceptor. The fact that buffering intracellular $Ca^{2+}$ by including BAPTA in the pipette solution eliminated the majority of fPSCs suggests that the ACR-14 autoreceptor is activated by acetylcholine release from the same neuron, and is likely located very close to the neurotransmitter release site. The inhibitory effects of *acr-14* mutation on spontaneous and evoked PSCs at the neuromuscular junction suggest that a physiological role of the ACR-14 autoreceptor in A-MNs is to facilitate neurotransmitter release. This role of ACR-14 autoreceptor in *C. elegans* cholinergic motor neurons is reminiscent of the functions of presynaptic AChRs in rat brain[55] and cultured *Xenopus* motor neurons[56].

Neuronal gap junctions are mainly known for their roles in synchronizing cellular activities[57–59]. The existence of morphologically or functionally mixed synapses in escape neural circuits

of various species[2,15,60] suggests that potential interactions between electrical and chemical synapses may play major roles in escape responses. This study shows that gap junctions allowing only antidromic current act to amplify chemical transmission between neurons with functionally mixed synapses. This finding resembles that of a recent study, which shows that electrical coupling may regulate chemical synaptic transmission at mixed synapses between premotor interneurons and motor neurons in zebrafish[5]. This similarity between two evolutionarily distant species suggests that amplification of chemical synapses by electrical synapses is likely a conserved mechanism in circuit functions. Why would nature design such a mechanism to amplify chemical transmission? Perhaps this mechanism is employed to sustain synaptic strength between two neurons without requiring more excitatory inputs into the presynaptic neuron, which is potentially beneficial to the function of diverse neural circuits. For example, a brief noxious stimulus could cause a much longer escape response because of the amplifying effect of chemical transmission by electrical coupling. In the mammalian

hippocampus, where mixed synapses exist[9,12,61], enhancement of chemical transmission by electrical coupling at mixed synapses could be a basic and necessary mechanism for learning and memory. Whether mixed synapses do play such physiological roles is an interesting question to be answered by future studies.

## Methods

**C. elegans culture and strains.** All worms were raised on agar plates with a layer of OP50 *Escherichia coli* at 21 °C inside an environmental chamber. The worm strains used in this study are listed in Table 1.

**Gene expression pattern analyses.** To confirm that *lgc-46* and *acr-14* are expressed in A-MNs, we coexpressed mStrawberry under the control of P*unc-4* (A-MN-specific) and GFP under the control of either P*lgc-46* or P*acr-14*. To confirm that AVA interneurons are cholinergic neurons, we first created an integrated transgenic strain expressing GFP under the control of P*unc-17* (ZW798), and then coinjected two plasmids, pNP259 (P*gpa-14*::Cre) and wp1392 (P*flp-18*::loxP::LacZ::STOP::loxP::mStrawberry) into the ZW798 strain. The use of P*gpa-14* and P*flp-18* in Cre-loxP recombination results in AVA-specific gene expression[30]. The expression patterns of GFP and mStrawberry were examined and imaged with an inverted microscope (TE-2000U, Nikon) equipped with specific fluorescence filters (HQ Texas Red/41004 and HQ FITC/41001, Chroma Technology Corp., Bellows Falls, VT, USA) and a CCD camera (F-View II, Olympus). P*flp-18* and P*gpa-14* were gifts from Alexander Gottschalk lab[30]. The other promoters, including P*unc-4* (2.9 kb), P*acr-14* (3.8 kb), P*lgc-46* (3.0 kb) and P*unc-17* (4.1 kb), were cloned using genomic DNA of the Bristol N2 strain as the template and the primers listed in Table 2.

**Rescue experiments.** *lgc-46* and *acr-14* mutants were rescued by expressing wild-type LGC-46 and ACR-14 under the control of either the native promoter or the A-MN-specific P*unc-4*. Full-length *lgc-46* (Y71D11A.5) cDNA and *acr-14* (T05C12.2) cDNA were cloned from a Bristol N2 cDNA library using the primers listed in Table 2.

**RNA interference.** Neuron-specific gene knockdown was achieved by coexpressing two plasmids encoding sense and corresponding antisense RNA fragment of a gene under the control of a specific promoter[62]. A-motor neuron-specific gene knockdown was achieved by using P*unc-4*, whereas AVA-specific gene knockdown through a Cre-LoxP approach using P*flp-18* and P*gpa-14* (ref. 30). The primers used for cloning the cDNA fragments of *lgc-46* (585 bp), *unc-17* (473 bp), *unc-7* (485 bp) and *unc-9* (397 bp) are listed in Table 2. One transgenic line of each kind was randomly chosen for electrophysiological and behavioural analyses.

**Auxin-induced UNC-7 degradation.** A DNA fragment encoding codon-optimized degron[41] was synthesized (GenScript, Piscataway, NJ, USA) and inserted into wild-type genome using the Crispr/Cas9 approach[42] to tag UNC-7 with degron at the C-terminus. The guide RNA sequence (5′-GGATGCGGAACA CGGTCAA) targeting the last exon of *unc-7* was inserted into pDD162 (P*eft-3*::Cas9 + Empty sgRNA) (Addgene #47549) by site-directed mutagenesis. The resulting plasmid (wp1685) was coinjected with the transgenic marker P*myo-2*::mStrawberry (wp1613) into wild-type worms. Genome-edited worms were identified through PCR screen and DNA sequencing. The Cre-LoxP system was used to achieve AVA-specific degradation of UNC-7. The TIR1::mRuby sequence was amplified from pLZ31 (P*eft-3*::TIR1::mRuby::*unc-54* 3′UTR, pCFJ151) (Addgene #71720) and cloned into P*flp-18*::loxP::LacZ::STOP::loxP::mStrawberry (wp1392) to generate P*flp-18*::loxP::LacZ::STOP::loxP::TIR1::mRuby (wp1693). This plasmid was coinjected with P*gpa-14*::Cre (pNP259) and P*flp-18*::loxP::LacZ:: STOP::loxP::mCherry::SL2::GFP (wp1383) into the UNC-7 degron strain. For auxin-induced UNC-7 degradation, L4 transgenic worms were transferred to NGM plates containing 1 mM auxin, and the adult worms were used for electrophysiological recording on the following day.

**C. elegans electrophysiology.** All electrophysiological experiments were performed with adult hermaphrodites. An animal was immobilized on a glass coverslip by applying Vetbond Tissue Adhesive (3M Company, St Paul, MN). Application of the glue was generally restricted to the dorsal anterior portion of the animal, allowing the tail to sway freely during the experiment. A short (~300 μm) longitudinal incision was made along the glued region. After clearing the viscera by suction through a glass pipette, the cuticle flap was folded back and glued to the coverslip, exposing several ventral body-wall muscle cells, a small number of motor neurons anterior to the vulva, and some neurons in the head. The dissected worm preparation was treated with collagenase A (Roche Applied Science, catalogue number 10103578001, 0.5 mg ml$^{-1}$) for 10–15 s and perfused with the extracellular solution for 5 to 10-fold of bath volume. Borosilicate glass pipettes

## Table 1 | List of worm strains.

| Strain ID | Genotype |
|---|---|
| N2 | Wild type (Bristol) |
| RB1132 | *acr-14(ok1155)* |
| PR1152 | *cha-1(p1152)* |
| RB2155 | *lgc-46(ok2900)* |
| VC2209 | *lgc-46(ok2949)* |
| CB933 | *unc-17(e245)* |
| CX14845 | *kyEx4863[Prig-3::HisCl1::SL2::mCherry(pNP471)]* |
| ZW495 | *zwIs132[Pmyo-3::GCaMP2(wp962), lin-15(+)]* |
| ZW582 | *zwIs134[Pglr-1::ChR2::YFP(wp693), Pmyo-2::DsRED(wp568)]* |
| ZW704 | *zwEx175[Pflp-18::loxP::LacZ::STOP::loxP::mCherry::SL2::GFP(wp1383), Pgpa-14::Cre(pNP259)].* |
| ZW725 | *zwEx176[Pflp-18::loxP::LacZ::STOP::loxP::unc-17ss(wp1470), Pflp-18::loxP::LacZ::STOP::loxP::unc-17as(wp1471), Pgpa-14::Cre(pNP259), Pmyo-2::YFP(wp214)].* |
| ZW731 | *zwEx175[Pflp-18::loxP::LacZ::STOP::loxP::mCherry::SL2::GFP(wp1383), Pgpa-14::Cre(pNP259)]; unc-7(e5)* |
| ZW732 | *zwEx175[Pflp-18::loxP::LacZ::STOP::loxP::mCherry::SL2::GFP(wp1383), Pgpa-14::Cre(pNP259)]; unc-9(fc16)* |
| ZW755 | *zwIs133[Punc-17::GFP(wp608)]; zwEx177[Pflp-18::loxP::LacZ::STOP::loxP::mStrawberry(wp1392), Pgpa-14::Cre(pNP259)]* |
| ZW782 | *zwEx178[Punc-4::lgc-46ss(wp1425), Punc-4::lgc-46as(wp1426), Pmyo-2::YFP(wp214)]* |
| ZW783 | *lgc-46(ok2949); zwEx179[Plgc-46::lgc-46(wp1462+wp1463), Pmyo-2::YFP(wp214)]* |
| ZW784 | *lgc-46(ok2900); zwEx179[Plgc-46::lgc-46(wp1462+wp1463), Pmyo-2::YFP(wp214)]* |
| ZW787 | *acr-14(ok1155); zwEx180[Punc-4::acr-14(wp1511), Pmyo-2::YFP(wp214)]* |
| ZW788 | *lgc-46(ok2900); zwEx181[Punc-4::lgc-46(wp1409), Pmyo-2::YFP(wp214)]* |
| ZW789 | *lgc-46(ok2949); zwEx181[Punc-4::lgc-46(wp1409), Pmyo-2::YFP(wp214)]* |
| ZW791 | *zwEx183[Punc-4::unc-17ss(wp1490), Punc-4::unc-17as(wp1491), Pmyo-2::YFP(wp214)]* |
| ZW800 | *zwEx188[Plgc-46::GFP(wp1366), Punc-4::mStrawberry(wp1400), lin-15(+)]* |
| ZW802 | *zwEx190[Pacr-14::GFP(wp1476), Punc-4::mStrawberry(wp1400), lin-15(+)]* |
| ZW933 | *zwEx175[Pflp-18::loxP::LacZ::STOP::loxP::mCherry::SL2::GFP(wp1383), Pgpa-14::Cre(pNP259)]; zwEx202[Pflp-18::loxP::LacZ::STOP::loxP::unc-7ss(wp1626), Pflp-18::loxP::LacZ::STOP::loxP::unc-7as(wp1627), Pgpa-14::Cre(pNP259), Pmyo-2:: mStrawerry(wp1613)]* |
| ZW934 | *zwEx175[Pflp-18::loxP::LacZ::STOP::loxP::mCherry::SL2::GFP(wp1383), Pgpa-14::Cre(pNP259)]; zwEx203[Punc-4::unc-9ss(wp1628), Punc-4::unc-9as(wp1629), Pmyo-2::mStrawerry(wp1613)]* |
| ZW972 | *zwEx175[Pflp-18::loxP::LacZ::STOP::loxP::mCherry::SL2::GFP(wp1383), Pgpa-14::Cre(pNP259)]; zwEx214[Pflp-18::loxP::LacZ::STOP::loxP::unc-17ss(wp1470), Pflp-18::loxP::LacZ::STOP::loxP::unc-17as(wp1471), Pgpa-14::Cre(pNP259), Pmyo-2::mStrawberry(wp1613)]* |
| ZW1034 | *unc-7::degron (zw87); zwEx218[Pflp-18::loxP::LacZ::STOP::loxP::TIR1::mRuby(wp1693), Pflp-18::loxP::LacZ::STOP::loxP::mCherry::SL2::GFP(wp1383), Pgpa-14::Cre(pNP259)]* |

**Table 2 | List of primers.**

| | Sense | Antisense |
|---|---|---|
| **Primers for cloning promoters** | | |
| P*unc-4* | 5′-ATGCCTGCAGCTCTGAAAATAT | 5′-CGCGGATCCTTTTCACTTTTTGGAAGA |
| P*acr-14* | 5′-CTTGGAAATGAAATAAGCTTTTGTTTGTAGATGAGCC | 5′-TCATTTTTTCTACCGGTACCTGATG ATGACCTCCTATTTGAA |
| P*lgc-46* | 5′-AAACTGCAGTACCCCAGGACTTTTCGTAG | 5′-ATCACCGGTAGCCCACCGTGCTCATCAACT |
| P*unc-17* | 5′-TTCGCATGCAAAACGGAGCTCGAGATTTT | 5′-ACTGTCGACTTCTACCGGTACCAATACGACTC |
| **Primers for cloning cDNAs** | | |
| *lgc-46* | 5′-GACGGATCCGCCACCATGCAATATCTGCAATTCCT | 5′-AAGCTCGAGTTAATAGCCCCGTAGATAGT |
| *acr-14* | 5′-TCAGGATCCGCCACCATGTCTTTTGTTTTCTTTATTTT | 5′-TTTCTAGATTAAATGTTTGCCAGTAT |
| **Primers for cloning cDNA fragments used in RNA interference** | | |
| *lgc-46* | 5′-ATGAACCCCTGTAAGTACAATTT | 5′-ATGTTACCAAACTGGAAAGTCAT |
| *unc-17* | 5′-ATGGATGATTTTGGGGCTATGC | 5′-ATGGTGGGCTCGAGGAAGGC |
| *unc-7* | 5′-GAACCATGGATGCTCGGCTCCTCCAGCA | 5′-TGTGCCGGCCATACTGCTTGGCGGAAACA |
| *unc-9* | 5′-GAGGATCCAGGATGAGTATGCTATTGTATT | 5′-ATTCTCGAGCTGATTGCCAGCTGAGCAG |

were used as electrodes for voltage- and current-clamp recordings. Pipette tip resistance for recording from neurons was $\sim 20 \, M\Omega$ whereas that for recording from body-wall muscle cells was $3-5 \, M\Omega$. Classical whole-cell configuration was obtained by applying a negative pressure to the recording pipette. Motor neurons were identified based on their anatomical locations whereas AVA based on GFP fluorescence from the expression of an integrated transgene, which was created using two promoters and the *Cre-LoxP* approach as described above, and crossed into various mutants. Spontaneous and evoked PSCs at the neuromuscular junction were recorded and analysed as previously described[63]. Voltage- and current-clamp experiments were performed with a Multiclamp 700B amplifier (Molecular Devices, Sunnyvale, CA, USA) and the Clampex software (version 10, Molecular Devices). Data were filtered at 2 kHz and sampled at 10 kHz. Postsynaptic currents and exogenous neurotransmitter/agonist-induced whole-cell currents were recorded at a holding voltage of $-60 \, mV$. Exogenous neurotransmitters and agonists were applied by pressure-ejection through a glass pipette using a Picospritzer III (Parker Hannifin, Hollis, NH) with pressure set at 10 pSi and pulse duration at 30 ms. The extracellular solution contained (in mM) NaCl 140, KCl 5, $CaCl_2$ 5, $MgCl_2$ 5, dextrose 11 and HEPES 5 (pH 7.2). The pipette solution contained (in mM) 120 KCl, 20 KOH, 5 Tris, 0.25 $CaCl_2$, 4 $MgCl_2$, 36 sucrose, 5 EGTA, and 4 $Na_2ATP$ (pH 7.2) except for that used for recording PSC bursts from body-wall muscle cells, which contained (in mM) 6.8 KCl, 113.2 Kgluconate, 20 KOH, 5 Tris, 0.25 $CaCl_2$, 4 $MgCl_2$, 36 sucrose, 5 EGTA and 4 $Na_2ATP$ (pH 7.2).

**Xenopus oocyte expression.** Oocytes were obtained from wild-type South African clawed frogs (Xenopus Express, Brooksville, FL) following a protocol approved by the Institutional Animal Care and Use Committee of UConn Health. Capped cRNA of *lgc-46* was synthesized using a T3 mMessage mMachine Kit (Life Technologies, Carlsbad, CA, USA), and injected into defolliculated oocytes ($\sim 50 \, nl$ per oocyte at $1.0 \, ng \, nl^{-1}$) using a Drummond Nanoject II injector (Drummond Scientific, Broomall, PA, USA). Two-electrode whole-cell recordings were performed 4–5 days after cRNA injection using an oocyte clamp amplifier (OC-725C, Warner Instruments, Hamden, CT, USA) and the Clampex software. The holding voltage was $-60 \, mV$. Data were filtered at 1 kHz and sampled at 10 kHz. In each experiment, a single oocyte was placed in a oocyte perfusion chamber (AutoMate Scientific, Berkeley, CA, USA) containing ND96 solution (compositions in mM): 96 NaCl, 2.0 KCl, 1.8 $CaCl_2$, 1.0 $MgCl_2$, 5.0 HEPES, pH 7.2). Candidate agonists were perfused into the oocyte perfusion chamber through a micro-manifold (AutoMate Scientific). An 8-channel perfusion controller (Valvelink8.2, AutoMate Scientific) was used for perfusion with its valves controlled by pClamp.

**$Ca^{2+}$ imaging.** Young adult worms of an integrated strain expressing GCaMP2 in body-wall muscle cells[54] were glued and filleted for imaging spontaneous fluorescence changes of body-wall muscle cells using an electron-multiplying CCD camera (iXonEM + 885, Andor Technology, Belfast, Northern Ireland), a FITC filter set (59222, Chroma Technology Corp.), a light source (Lambda XL, Sutter Instrument, Novato, CA, USA), and the NIS-Elements software (Nikon). Images were acquired at 16 frames per second with 10–40 ms of exposure time (no binning) for 3 min. TTL signals from the camera were used to synchronize the recordings of $Ca^{2+}$ transients and PSCs.

**Optogenetic stimulation.** Worms of an integrated strain expressing channelrhodopsin-2 in command interneurons under the control of P*glr-1* (ref. 16) were grown to L1-L2 stage on standard worm culture plates, and then transferred to new plates either with or without (for negative control) all-trans retinal two days before the experiment. The retinal plates were prepared by spotting each plate

(60-mm diameter with 10 ml agar) with 200 μl OP50 *E. coli* containing 2 mM retinal (R2500, Sigma-Aldrich). Photostimulation was applied through a 40X water immersion objective in 2-s pulses at 30-s intervals using a light source (Lambda XL with *Smart*Shutter, Sutter Instrument) and a $470 \pm 20 \, nm$ excitation filter (59222, Chroma Technology Corp.). The measured light intensity at the specimen position was $6.7 \, mW \, mm^{-2}$, which was sufficient to cause maximal evoked peak responses[16].

**Measurements of backward locomotion.** Backward locomotion in response to head touch by a platinum wire was assayed with L4-stage worms. The number of dorsoventral tail swings in response to each touch was counted by the experimenter with a stereomicroscope (SMZ-800, Nikon). One investigator (B.C.) prepared the worm strains whereas another (P.L.) performed the locomotion assay without knowing the strain genotypes.

**Chemicals.** Acetylcholine (AC159170050, ACROS Organics), GABA (AC103280250, ACROS Organics), aspartate (11240, Sigma-Aldrich), ATP (A3377, Sigma-Aldrich), dopamine (H8502, Sigma-Aldrich), glutamate (49621, Sigma-Aldrich), octopamine (O0250, Sigma-Aldrich), serotonin (H9523, Sigma-Aldrich), tyramine (T2879, Sigma-Aldrich), nicotine (AC181420050, ACROS Organics), levamisole (AC187870100, ACROS Organics), choline (AC110295000, ACROS Organics), glycine (G46-1, Fisher Scientific), d-tubocurarine (T2397, Sigma-Aldrich), gabazine (S106, Sigma-Aldrich), and histamine (H7250, Sigma-Aldrich) were first dissolved in water to make frozen stock solutions (10 to 100 mM), which were diluted to final concentrations using extracellular solutions before use.

**Data analyses.** The duration and charge transfer of PSC bursts were quantified with Clampfit software (version 10, Molecular Devices) as previously described[16]. The frequency of PSC bursts was manually counted. Amplitudes of junctional current and membrane voltage were quantified based on the mean amplitude during the last 100 ms at each voltage step (250 ms) or current injection step (1,000 ms) using pClamp. Calcium imaging data were analysed as previously described[54]. Data graphing and statistical analyses were performed with Origin Pro 2015 (OriginLab Corporation, Northampton, MA). Data are shown as mean $\pm$ s.e. Either ANOVA (with Tukey's HSD test) or unpaired $t$-test was used for statistical comparisons as specified in figure legends. $P < 0.05$ is considered to be statistically significant. The sample size ($n$) equals to the number of cells or cell pairs recorded, or the number of worms used in locomotion analyses.

**Data availability.** All data generated or analysed during this study are included in this published article and its Supplementary Information files.

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

## Acknowledgements

This work was supported by National Institute of Health (2R01MH085927 to Z.-W.W., and R01GM113004 to B.C.) and Air Force Office of Scientific Research (FA9550-15-1-0060 to R.M.). We thank Cori Bargmann for her insightful suggestions during the course of this study and for the HisCl1 strain. We thank Alexander Gottschalk for the pCoS10 and pNP259 plasmids, Abby F. Dernburg for the pLZ31 plasmid, Xiaoming Xia for an oocyte expression vector, and Caenorhabditis Genetics Center (USA) for mutant strains.

## Author contributions

P.L. did all the electrophysiological experiments and data analyses. B.C. did the molecular biology experiments, created the new transgenic strains, and analysed gene expression patterns. P.L., Z.-W.W., B.C. and R.M. designed the experiments. P.L., Z.-W.W. and B.C. wrote the manuscript.

## Additional information

**Competing interests:** The authors have no competing interests as defined by Nature Publishing Group, or other interests that might be perceived to influence the results and/or discussion reported in this article.

**Publisher's note**: 

