## [Peer Review File · Nature Communications]

Reviewers' comments:

Reviewer #1 (Remarks to the Author):

This study uses a range of techniques, in particular optogenetics and electrophysiology, to investigate the motor circuit of *C. elegans*. The authors obtain a number of interesting findings, most notably discovering that mixed synapses combining excitatory cholinergic chemical synapses and rectifying gap junctions produce sustained activity that is crucial to the escape response in the worm. They also identify several other key molecular players in the circuit, including the acetylcholine receptors LGC-46 and ACR-14, and discern their cellular sites of action.

Overall, this is a very nice paper, dense with experimental detail and yielding important mechanistic insight into behavior. I am supportive of publication in *Nature Communications*, and have only minor suggestions for revision.

The main experimental query I have concerns the cell-specific RNAi knockdown of genes such as *unc-17*. Given the possibility that RNAi triggers can spread between cells, even neurons, it would be good to include controls to show that effect is really cell-specific. I.e., in the AVA-specific line, is there a way to show that cholinergic transmission in other neurons is unaffected?

Also, in the discussion the authors argue against the conclusions of Kawano et al (that the uncoordination of *unc-7* mutants is due to shunting through gap junctions) since this seems inconsistent with the antidromic rectification they observe in their study. Their alternative explanation of Kawano et al's results was rather brief though. Could they possibly elaborate a bit more here?

Reviewer #2 (Remarks to the Author):

This paper "Antidromic-rectifying gap junctions amplify chemical transmission at a mixed electrical chemical synapse" by Liu et al. represents an excellent contribution to our understanding of interactions between chemical and electrical transmission. The paper describes the properties of synaptic communication between the AVA cell (a premotor interneuron) and motoneurons (A-MNs) that are part of an escape circuit in *C. elegans*. These neurons are communicated via chemical and electrical synapses and both forms of transmission were shown to be required for the escape response. Chemical synapses mediate anterograde transmission (AVA to MNS) and is mediated by acetylcholine. On the other hand, electrical transmission mediates anterograde transmission (MNs to AVA) via strongly rectifying channels that allow depolarizations to flow only in this direction. Such arrangement results in enhancement of chemical transmission, which the authors argue is functionally relevant for the escape network. The conclusions are supported by electrophysiological recordings in combination with pharmacology and genetic tools. Given the small size of the *c. elegans*, the electrophysiological recordings represent a technical tour de force.

The paper is nicely written and clearly illustrated (I have a lot of fun reading it). There are on the other hand a number of concerns that need to be addressed:

- The authors define this transmission as a "mixed synapse" (i.e., like those in goldfish, zebrafish and hippocampus). What is the morphological evidence for this? This evidence should be discussed/provided. Alternatively, they could be two independent reciprocal synapses: one chemical from AVA to MN and one electrical from MN to AVA. Interactions would be heterosynaptic in this case.

- Line 218: "Electrical coupling between AVA and A-MNs only allows antidromic current and requires

UNC-7 and UNC-9 innexins” This conclusion is confusing to me. Unless I am missing something, according to Fig. 6b current seems to flow in both directions (inward from VA5 to AVA and outward from AVA to VA5). So, depending on the polarity of V_j current can go in both directions. This is consistent with rectification being a voltage-dependent property and the I/V relationships obtained by the authors resemble very much those found in crayfish (see Fig. 13 in Furshpan and Potter, 1959). What it seems to be unidirectional is electrical transmission, which is influenced by both GJ conductance and the input resistance of the cells. Consistent with Fig. 6b, depolarization of VA5 in Fig. 6c clearly favors transmission of depolarizations over hyperpolarizations like from Pre to Post in crayfish). According to Fig. 6B, presynaptic (AVA) hyperpolarization should cause a postsynaptic hyperpolarizing coupling potential, but no change in membrane potential was detected. This could result from differences between the input resistance of the coupled cells. In other words, unlike that of AVA, the input resistance of VA5 could be too low for the GJ current to produce any detectable change in the membrane potential. If so, transmission could be exposed by including K^+ channel blockers in VA5 to increase its input resistance. Could this be the case? Did the authors measure the input resistance of each cell? This issue should be clarified and current flow vs electrical transmission (expressed as coupling coefficient) disambiguated.

- Line 283: “In this study, we showed that the electrical coupling between AVA and A-MNs allows current flow only in one direction, which is probably the strongest degree of rectification ever observed with gap junctions in native tissues.” See above. Current seems here to be bidirectional, but electrical transmission (coupling coefficient) seems to be unidirectional. All rectifying synapses are bi-directional but they are polarized. Asymmetry of electrical transmission simply result from strong differences between the input resistances of the coupled cells, not of GJ conductance.

- Line 273: “Our study also shows that motoneurons are not simply passive responders to commands but may modulate the activities of higher order neurons retrogradely.” This conclusion was also reached by Song et al. (Nature, 2016) at zebrafish mixed synapses. Reference to this paper will greatly enhance the results shown here, as it demonstrates that this is a robust, converging, property across different nervous systems. Your results are sufficiently different and in any way affects the novelty of the results provided in this article.

Minor concerns:

- Line 37: “At a mixed synapse between auditory afferents and Mauthner cells (interneurons) in goldfish,” The Mauthner cell is not generally described as an interneuron, it is a cell that forms part of the reticulospinal system.

- Line 103: “Excellent temporal correlations were observed between PSC bursts”. What is an “excellent” temporal correlation? Please define or modify.

- Fig. 3A: It would be nice to show a magnified version of the response to optogenetic stimulation during TBC. Can small electrically mediated responses be detected?

- Line 221: “We found that gap junctions at this synapse are strongly rectified in the AVA direction” It should read strongly RECTIFYING.

- Line 271: “Our work with *C. elegans* shows that antidromic rectifying gap junctions can facilitate orthodromic chemical transmission at a mixed synapse,...” Facilitation of chemical release by retrograde electrical transmission at mixed synapses was also proposed to occur at goldfish mixed synapses (see Fig. 4 in Pereda et al., Brain Research Reviews, 2004).

Reviewer #3 (Remarks to the Author):

This study by Liu et al. focuses on the exciting topic of how the electrical and chemical components of a mixed synapse interact, as well as their effects on circuit function and behavior. The authors use as a model a specific synapse in the *C.elegans* motor circuit and carefully perform current- and voltage-clamp recordings which lead them to conclude that gap junctions serve as amplifiers of chemical transmission at a mixed synapse. The results presented here are quite exciting but several conclusions need to be strengthened by additional experiments and clarifications as suggested below.

Major points:

1. What is the evidence that the AVA->VA5 synapse is actually a mixed synapse? Are there any electron microscopy data or other sorts of data that support this conclusion? If yes, this should be mentioned in the text and included in the results. If it is indeed a mixed synapse, are the AVA-VA5 gap junctions adjacent to the AVA-VA5 active zones? In the WormWiring website, VA5 chemical and electrical synapses with AVA seem "far away" from each other. They authors must clearly define what do they mean by the term "mixed synapse" in the Introduction.
2. The authors specifically study the AVA-VA5 synapse because VA5 is easily identifiable. Based on their findings they generalize that their findings apply to all A-type motor neurons, which include the VA and DA class. If they insist on this claim, then some data on any of the AVA-DA synapses must be included in the manuscript, especially in light of the fact that there is solid evidence that the DA motor neurons are required for backward locomotion.
3. The authors have gone to a great length to carefully perform SPONTANEOUS current and voltage-clamp recordings. In the Methods section, it is described that the animal is immobilized with the use of tissue adhesive but the rest of its body/tail can still move. So, my question is: Since the authors observe 2 distinct populations of PSCs, can it be that the worm is performing 2 distinct types of movement (e.g. dorsal bends, ventral bends) while these recordings are taken? In other words, can we associate these 2 types of PSCs to 2 distinct behaviors?
4. The authors discuss that their results are contradictory to what described in Kawano et al. 2012. They attribute these differences to the ectopic gap junctions of AVA and B-type motor neurons observed in EM data of *unc-7* mutants. Since the RNAi knockdown of *unc-7* is done using an AVA-specific promoter, which is continuously active (from early development to adult), it is possible that ectopic synapses between AVA and B-type MNs are also formed in the AVA-specific *unc-7* RNAi knockdown. Removal of *unc-7* post-developmentally using the auxin-inducible system (Zhang et al., Development, 2015) could clarify this point. Also, the authors study VA5 while Kawano et al study VA8. As VA5 and VA8 are located in two different locations of the ventral nerve cord, they may serve different functions. This point could be further discussed and perhaps strengthened if the authors also get some VA8 recordings. In addition, the conclusions drawn from figure 7 could be strengthened by performing rescue experiments (e.g. provide *unc-7* in AVA of *unc-7* mutants).
5. It is not clear if TWO blind observers to the genotypes tested performed the measurements for backward locomotion. From the Methods section, it appears that one observer performed the experiment.
6. The authors do not show/state how many transgenic lines have been used to knockdown *unc-7* and *unc-9* in AVA and A-type MNs respectively. This information is critical and additional data must be shown in Fig 7C and Fig. 8A. Also, the AVA knock down shown on these figures was achieved with *flp-*

18 or gpa-4 promoters or both?

Minor points:

1. To avoid confusion, the term "A-class motor neurons" should be changed to as "A-type motor neurons". The term class in *C.elegans* ventral nerve cord has been reserved for DA, DB, VA, VB, AS motor neurons.
2. When different exogenous neurotransmitters are applied to VA5, there is an effect with ACh and GABA. The authors demonstrate well the role of ACh in AVA-VA5 synapse but they fail to mention (or speculate) why GABA also induced a current.
3. When describing lgc-46 and acr-14, the authors should mention which other lgc and acr mutants have examined. It is clear why they picked lgc-46 and acr-14, but not clear how many other mutants they examined.
4. The mechanism of action for BAPTA must be described. The authors take for granted that the reader knows a priori what BAPTA does.
5. Which exact muscle is innervated by VA5?

Reviewer #1

- *The main experimental query I have concerns the cell-specific RNAi knockdown of genes such as unc-17. Given the possibility that RNAi triggers can spread between cells, even neurons, it would be good to include controls to show that effect is really cell-specific. I.e., in the AVA-specific line, is there a way to show that cholinergic transmission in other neurons is unaffected?*

Thanks for this excellent comment. In the manuscript, we have results showing that AVA-specific unc-17 RNAi inhibited sPSCs (depending on acetylcholine release from AVA) but not fPSCs (depending on acetylcholine release from A-MNs) in VA5 (Fig. 3), suggesting that RNAi in AVA did not spread to A-MNs. We also have results showing that A-MN-specific unc-17 RNAi inhibited VA5 fPSCs but not sPSCs (Fig. 5), suggesting that A-MN-specific RNAi did not spread to AVA. However, the original manuscript did not relate these results to address the concern over potential off-target effects, which is now addressed (pages 5, 3rd paragraph and page 6, 4th paragraph).

- *Also, in the discussion the authors argue against the conclusions of Kawano et al (that the uncoordination of unc-7 mutants is due to shunting through gap junctions) since this seems inconsistent with the antidromic rectification they observe in their study. Their alternative explanation of Kawano et al's results was rather brief though. Could they possibly elaborate a bit more here?*

Our conclusion that antidromic-rectifying gap junctions between AVA and A-MNs amplify chemical transmission is reinforced by several additional experiments, including analyzing electrical coupling and chemical transmission between AVA and additional A-type motor neurons (DA4 and VA8, Supplementary Figs. S3, S5), and AVA-specific removal of UNC-7 in adult worms using an auxin-inducible system (Supplementary Fig. S4). As suggested by this reviewer, the revised manuscript contains more discussions about potential causes for the different conclusion of the previous study (page 9, last paragraph).

Reviewer #2

- *The authors define this transmission as a “mixed synapse” (i.e., like those in goldfish, zebrafish and hippocampus). What is the morphological evidence for this? This evidence should be discussed/provided. Alternatively, they could be two independent reciprocal synapses: one chemical from AVA to MN and one electrical from MN to AVA. Interactions would be heterosynaptic in this case.*

The chemical and electrical synapses between AVA and A-MNs are not closely associated like those morphologically mixed synapses in goldfish, zebrafish and hippocampus. Nevertheless, they can functionally interact because *C. elegans* neurons generally have a very high membrane resistance, and changes of membrane voltage are instantaneously spread to the entire neuron. These properties of *C. elegans* neurons are now described in the Introduction (page 3, 3rd paragraph).

We used the word “mixed synapse” because it is sometimes used to refer to a functional unit or synapses showing interactions between electrical and chemical synapses without morphological data (e. g. Mamiya, et al., J Neurosci 23: 9557-9564, 2003). During the revision of this manuscript, we consulted with Dr. Alberto Pereda (the author of Reference #1) about the most appropriate words to describe the synapses between AVA and A-MNs. He told us that the definition of mixed synapse is something that he often discusses and argues with his colleagues. He thinks that it may be better to describe the synapses between AVA and A-MNs as “functionally mixed synapses” than “mixed synapses” to distinguish them from morphologically mixed synapses. Although the electrical and chemical synapses between AVA and A-MNs are not closely associated in structure, knowledge from these synapses is potentially very important to understanding how chemical and electrical synapses might interact at morphologically mixed synapses.

The synapses between AVA and A-MNs are essentially like “two independent reciprocal synapses” in anatomy, as suggested by the reviewer. We are uncertain whether it is appropriate to use the word “heterosynaptic” here. Based on our understanding, the word “heterosynaptic” is generally used to describe interactions involving at least three neurons. For example, the regulation of gap junctions between retinal amacrine cells by chemical synaptic input from bipolar cells (the third neuron) is considered to be heterosynaptic (please see reference #1). In our case, the interactions between chemical and electrical synapses do not involve a third neuron. Please correct us if we did not understand the meaning of “heterosynaptic” correctly.

- *Line 218: “Electrical coupling between AVA and A-MNs only allows antidromic current and requires UNC-7 and UNC-9 innexins” This conclusion is confusing to me. Unless I am missing something, according to Fig. 6b current seems to flow in both directions (inward from VA5 to AVA and outward from AVA to VA5). So, depending on the polarity of V_j, current can go in both directions. This is consistent with rectification*

being a voltage-dependent property and the I/V relationships obtained by the authors resemble very much those found in crayfish (see Fig. 13 in Furshpan and Potter, 1959). What it seems to be unidirectional is electrical transmission, which is influenced by both GJ conductance and the input resistance of the cells. Consistent with Fig. 6b, depolarization of VA5 in Fig. 6c clearly favors transmission of depolarizations over hyperpolarizations like from Pre to Post in crayfish). According to Fig. 6B, presynaptic (AVA) hyperpolarization should cause a postsynaptic hyperpolarizing coupling potential, but no change in membrane potential was detected. This could result from differences between the input resistance of the coupled cells. In other words, unlike that of AVA, the input resistance of VA5 could be too low for the GJ current to produce any detectable change in the membrane potential. If so, transmission could be exposed by including K⁺ channel blockers in VA5 to increase its input resistance. Could this be the case? Did the authors measure the input resistance of each cell? This issue should be clarified and current flow vs electrical transmission (expressed as coupling coefficient) disambiguated.

We apologize for the confusion. Fig. 6b shows that junctional current flows in only one direction (from VA5 to AVA), which is an intrinsic property of the heterotypic gap junctions formed by two different innexins (UNC-7 in AVA and UNC-9 in A-MNs). Although both depolarizing and hyperpolarizing junctional voltage steps were applied, only inward I_j were observed in AVA and only outward I_j were observed in VA5 (Fig. 6b). As suggested, we measured the input resistance of AVA and VA5. Both AVA and VA5 have very high input resistance ($2.50 \pm 0.19 \text{ G}\Omega$ and $3.47 \pm 0.18 \text{ G}\Omega$, respectively), which is now described in the manuscript (page 8, 5th paragraph). We feel that a poorly phrased sentence in the original manuscript had contributed to the confusion. The sentence was: "Significant junctional current (I_j) was observed only at depolarizing junctional voltage (V_j) steps applied to VA5 but only at hyperpolarizing V_j steps applied to AVA (Fig. 6b), indicating that I_j flows only in the AVA direction." It is now replaced by the following sentence: "Significant junctional current (I_j) was observed only when the membrane voltage (V_m) of VA5 was more depolarized than that of AVA (Fig. 6b), indicating that I_j flows only in the AVA direction." In addition, we made some changes to the figure legend. Hopefully, the reviewer will find that the revised description no longer confusing.

- *Line 283: "In this study, we showed that the electrical coupling between AVA and A-MNs allows current flow only in one direction, which is probably the strongest degree of rectification ever observed with gap junctions in native tissues." See above. Current seems here to be bidirectional, but electrical transmission (coupling coefficient) seems to be unidirectional. All rectifying synapses are bi-directional but they are polarized. Asymmetry of electrical transmission simply result from strong differences between the input resistances of the coupled cells, not of GJ conductance.*

This comment is related to the last one. As we explained above, junctional current flows only in the VA5 to AVA direction, which is an intrinsic property of the gap junctions. It

seems that the unidirectional “electrical transmission” is not due to a large difference in input resistance of the coupled cells but due to the intrinsic property of the gap junctions.

- *Line 273: “Our study also shows that motoneurons are not simply passive responders to commands but may modulate the activities of higher order neurons retrogradely.” This conclusion was also reached by Song et al. (Nature, 2016) at zebrafish mixed synapses. Reference to this paper will greatly enhance the results shown here, as it demonstrates that this is a robust, converging, property across different nervous systems. Your results are sufficiently different and in any way affects the novelty of the results provided in this article.*

We thank the reviewer for this excellent suggestion. The Discussion section has been revised accordingly (page 11, 1st paragraph).

Minor concerns:

- *Line 37: “At a mixed synapse between auditory afferents and Mauthner cells (interneurons) in goldfish,” The Mauthner cell is not generally described as an interneuron, it is a cell that forms part of the reticulospinal system.*

We have corrected this error.

- *Line 103: “Excellent temporal correlations were observed between PSC bursts”. What is an “excellent” temporal correlation? Please define or modify.*

We have modified the sentence. The new sentence is: “Both PSC bursts and Ca²⁺ transients in muscle cells are temporally correlated with PSC bursts in VA5” (page 4, 3rd paragraph).

- *Fig. 3A: It would be nice to show a magnified version of the response to optogenetic stimulation during TBC. Can small electrically mediated responses be detected?*

It is now shown in Fig. 3a. The magnified trace still does not reveal any detectable electrically mediated responses.

- *Line 221: “We found that gap junctions at this synapse are strongly rectified in the AVA direction” It should read strongly RECTIFYING.*

Corrected.

- *Line 271: “Our work with C. elegans shows that antidromic rectifying gap junctions can facilitate orthodromic chemical transmission at a mixed synapse, ...” Facilitation of chemical release by retrograde electrical transmission at mixed synapses was also*

proposed to occur at goldfish mixed synapses (see Fig. 4 in Pereda et al., Brain Research Reviews, 2004).

Thanks for this suggestion. We have added this review article after the following sentence in the Introduction: “It has been proposed that the antidromic junctional current may promote cooperativity among afferents.” (page 3, 2nd paragraph).

Reviewer #3

Major points:

1. What is the evidence that the AVA->VA5 synapse is actually a mixed synapse? Are there any electron microscopy data or other sorts of data that support this conclusion? If yes, this should be mentioned in the text and included in the results. If it is indeed a mixed synapse, are the AVA-VA5 gap junctions adjacent to the AVA-VA5 active zones? In the WormWiring website, VA5 chemical and electrical synapses with AVA seem “far away” from each other. They authors must clearly define what do they mean by the term “mixed synapse” in the Introduction.

As pointed out by this reviewer, the electrical and chemical synapses between AVA and VA5 are not closely associated. Nevertheless, they can functionally interact because any voltage change is instantaneously spread to the entire neuron in these neurons due to a very high membrane resistance. During the revision of this manuscript, we consulted with Dr. Alberto Pereda (the author of Reference #1) about the most appropriate words to describe the synapses between AVA and A-MNs. He told us that the definition of mixed synapse is something that he often discusses and argues with his colleagues. He thinks that it may be better to describe the synapses between AVA and A-MNs as “functionally mixed synapses” than “mixed synapses” to distinguish them from morphologically mixed synapses. In the revised manuscript, the electrical and chemical synapses between AVA and A-MNs are no longer simply referred to as “mixed synapses”. We defined them as functionally mixed synapses in the last paragraph of the Introduction.

2. The authors specifically study the AVA-VA5 synapse because VA5 is easily identifiable. Based on their findings they generalize that their findings apply to all A-type motor neurons, which include the VA and DA class. If they insist on this claim, then some data on any of the AVA-DA synapses must be included in the manuscript, especially in light of the fact that there is solid evidence that the DA motor neurons are required for backward locomotion.

Thanks for this excellent suggestion. The manuscript now has data from DA4, showing that it behaves like VA5 in electrical coupling with AVA and in interactions with AVA through electrical and chemical synapses (Supplementary Fig. S5).

3. The authors have gone to a great length to carefully perform SPONTANEOUS current and voltage-clamp recordings. In the Methods section, it is described that the animal is immobilized with the use of tissue adhesive but the rest of its body/tail can still move. So, my question is: Since the authors observe 2 distinct populations of PSCs, can it be that the worm is performing 2 distinct types of movement (e.g. dorsal bends, ventral bends) while these recordings are taken? In other words, can we associate these 2 types of PSCs to 2 distinct behaviors?

It seems unlikely that the two different types of PSCs are associated with two different behaviors because they occur in a mixed manner no matter how the immobilized worm bends its head or tail. Our results indicate that sPSCs result from chemical synaptic transmission from AVA and are important to backward locomotion, whereas fPSCs result from the activation of an autoreceptor and have an enhancing effect on neurotransmitter release from motor neurons.

4. The authors discuss that their results are contradictory to what described in Kawano et al. 2012. They attribute these differences to the ectopic gap junctions of AVA and B-type motor neurons observed in EM data of unc-7 mutants. Since the RNAi knockdown of unc-7 is done using an AVA-specific promoter, which is continuously active (from early development to adult), it is possible that ectopic synapses between AVA and B-type MNs are also formed in the AVA-specific unc-7 RNAi knockdown. Removal of unc-7 post-developmentally using the auxin-inducible system (Zhang et al., Development, 2015) could clarify this point. Also, the authors study VA5 while Kawano et al study VA8. As VA5 and VA8 are located in two different locations of the ventral nerve cord, they may serve different functions. This point could be further discussed and perhaps strengthened if the authors also get some VA8 recordings. In addition, the conclusions drawn from figure 7 could be strengthened by performing rescue experiments (e.g. provide unc-7 in AVA of unc-7 mutants).

We performed three different experiments to address these very thoughtful comments: 1) we used the auxin-inducible system to remove UNC-7 specifically in AVA in adult worms, and obtained similar results as AVA-specific unc-7 RNAi (Supplementary Fig. S4); 2) we analyzed electrical coupling between AVA and VA8, and found that junctional currents are also only in the antidromic direction (Supplementary Fig. S3); and 3) we analyzed electrical coupling between AVA and VB6 in both wild type and AVA-specific unc-7 knockdown worms, and did not observe any junctional current (Supplementary Fig. S7). These results suggest that our results and interpretations are unlikely complicated by either ectopic gap junctions or developmental changes, and that the results obtained with VA5 are representative of A-MNs (the revised manuscript also includes results from DA4, as described above).

5. It is not clear if TWO blind observers to the genotypes tested performed the

measurements for backward locomotion. From the Methods section, it appears that one observer performed the experiment.

One investigator (BC) prepared the worm strains whereas another (PL) performed the locomotion assay without knowing the strain genotypes. This is now described in the manuscript (page 15, 1st paragraph).

6. The authors do not show/state how many transgenic lines have been used to knockdown unc-7 and unc-9 in AVA and A-type MNs respectively. This information is critical and additional data must be shown in Fig 7C and Fig. 8A. Also, the AVA knock down shown on these figures was achieved with flp-18 or gpa-4 promoters or both?

Although multiple transgenic lines were generated for unc-7 and unc-9 knockdown, we randomly picked one line from each knockdown for our experiments, which is now described in the manuscript (page 13, 1st paragraph). We did not perform analysis on additional transgenic lines because the data obtained with the selected lines show that AVA-specific unc-7 RNAi and A-MN-specific unc-9 RNAi have similar effects on synaptic transmission and the escape response, which is in agreement with the previous finding that UNC-7 in AVA and UNC-9 in A-MNs form heterotypic gap junctions. Additionally, our new experiments show that electrical and chemical transmissions between AVA and DA4 are similarly inhibited in the AVA-specific unc-7 knockdown strain (Supplementary Fig. S5), and that AVA-specific removal of UNC-7 in adult worms using the auxin-inducible system produces similar results as AVA-specific unc-7 knockdown (Supplementary Fig. S4). AVA-specific knockdown was achieved using the Cre-LoxP approach with both flp-18 and gpa-14 promoters, as described in the manuscript (page 5, 3rd paragraph, page 12, last paragraph, and page 13, 2nd paragraph).

Minor points:

1. To avoid confusion, the term “A-class motor neurons” should be changed to as “A-type motor neurons”. The term class in C. elegans ventral nerve cord has been reserved for DA, DB, VA, VB, AS motor neurons.

Thanks for pointing it out. We have corrected the error.

2. When different exogenous neurotransmitters are applied to VA5, there is an effect with ACh and GABA. The authors demonstrate well the role of ACh in AVA-VA5 synapse but they fail to mention (or speculate) why GABA also induced a current.

The observation suggests that there is also a GABA receptor in VA5, which is now described in the manuscript (page 4, last paragraph). GABA motoneurons might be the source of neurotransmitter for activating the GABA receptor because the worm

connectome indicates that there are chemical synapses from GABA motoneurons to acetylcholine motoneurons (<http://wormweb.org/neuralnet#c=VA&m=1>).

*3. When describing *lgc-46* and *acr-14*, the authors should mention which other *lgc* and *acr* mutants have examined. It is clear why they picked *lgc-46* and *acr-14*, but not clear how many other mutants they examined.*

We selected candidate genes that are known to be expressed in motoneurons but are not components of a previously identified motoneuron extrasynaptic receptor. This information is now in the manuscript (page 5, last paragraph).

4. The mechanism of action for BAPTA must be described. The authors take for granted that the reader knows a priori what BAPTA does.

Thanks for pointing this out. The manuscript has been revised to make it clearer why BAPTA was used in the experiment (page 6, last paragraph).

5. Which exact muscle is innervated by VA5?

This information is currently unavailable in published literatures. The laboratories of Drs. David Hall and Scott Emmons are currently quantifying all synapses in *C. elegans* using several approaches, and plan to submit a manuscript in a few months. According to Dr. Hall (personal communication), VA5 has neuromuscular junctions on the left side to ventral body-wall muscle cells [UNPUBLISHED DATA REDACTED], and on the right side to ventral body-wall muscle cells [UNPUBLISHED DATA REDACTED]. They prefer to keep this information confidential at this time.

REVIEWERS' COMMENTS:

Reviewer #1 (Remarks to the Author):

I think the authors have done a good job addressing reviewer comments. This is an interesting and thorough manuscript and I am supportive of publication.

Reviewer #2 (Remarks to the Author):

The authors have addressed the reviewer's concerns.

Reviewer #3 (Remarks to the Author):

None.